

# Effects of High-Quality Elevation Data and Explanatory Variables on the Accuracy of Flood Inundation Mapping via Height Above Nearest Drainage

Fernando Aristizabal[1,2,3], Taher Chegini[4], Gregory Petrochenkov[1,2], Fernando Salas[2], and Jasmeet Judge[3]

[1]Lynker, 338 E Market St, Leesburg, VA, 20176, USA
[2]National Water Center, Office of Water Prediction, National Oceanic and Atmospheric Administration, 205 Hackberry Ln, Tuscaloosa, AL, 35401, USA
[3]Center for Remote Sensing, Agricultural and Biological Engineering, University of Florida, 1741 Museum Rd, Gainesville, FL, 32603, USA
[4]Civil and Environmental Engineering, University of Houston, 4226 Martin Luther King Boulevard, Houston, TX, 77204, USA

**Correspondence:** Fernando Aristizabal (fernando.aristizabal@noaa.gov)

**Abstract.** Given the availability of high quality and high spatial resolution digital elevation models (DEMs) from the United States Geological Survey's 3-Dimensional Elevation Program (3DEP) derived from mostly Light Detection and Ranging sensors, we examined the effects of these DEMs at various spatial resolutions on the quality of flood inundation map (FIM) extents derived from a terrain index known as Height Above Nearest Drainage (HAND). We found that using these DEMs improved the quality of resulting FIMs at around 80% of the catchments analyzed when compared to using DEMs from the National Hydrography Dataset Plus High Resolution program. Additionally, we varied the spatial resolution of the 3DEP DEMs from 3, 5, 10, 15, and 20 meters and the results showed no significant overall effect on FIM extent quality across resolutions. However, our experiments demonstrated a significant burden on the computational time to produce HAND. We fit a multiple linear regression model to help explain catchment scale variation in the four metrics employed and found that the lack of reservoir flooding, or inundation upstream of river retention systems, was a significant factor in our analysis. For validation, we used Interagency Flood Risk Management Base Level Engineering produced FIM extents and streamflows at the 100 and 500 year event magnitudes in a sub-region in Eastern Texas.

## 1 Introduction

Floods are among the most frequent, damaging, and deadly of natural disasters (Doocy et al., 2013; Strömberg, 2007; Kahn, 2005). The frequency and intensity of flood events as well as the exposure of people and property to them have been increasing in recent times driven by secular changes in climate, infrastructure, and demographics (Berz, 2000; Mallakpour and Villarini, 2015; Downton et al., 2005; Kunkel et al., 1999; Pielke Jr and Downton, 2000; Corringham and Cayan, 2019; Gourevitch et al., 2023). These upward trends are expected to continue placing additional pressure on hydrological extremes (Kahn, 2005; Tabari, 2020; Milly et al., 2002; Wing et al., 2018; Gourevitch et al., 2023). Floods impact mortality and morbidity through



drowning or physical trauma at the individual health scale, while increasing the risk of infectious disease at the public health
level (Jonkman, 2005; Beinin, 2012; Alajo et al., 2006; French et al., 1983). Flooding disrupts systems providing human needs
such as transportation routes, supply chains, water delivery, waste management, communications, shelter, and energy grids
(Wijkman and Timberlake, 2021; Gourevitch et al., 2023). These impacts disproportionately affect certain demographics such
as the socioeconomically-disadvantaged, youth, and elderly who are more likely to live in vulnerable areas with less access to
educational resources, early warning systems, and the capacity or resources to evacuate impacted areas (Kahn, 2005; Smiley
et al., 2022; Strömberg, 2007; Jonkman, 2005; Tellman et al., 2020, 2021). These inequitable impacts further entrench poverty
and inequalities (Stallings, 1988; Birkmann et al., 2010). In political terms, severe disasters, including floods, can reduce social
order, strain governance systems, collapse social safety nets, and increase the risk of social conflict (Drury and Olson, 1998;
Xu et al., 2016; Zahran et al., 2009). These dire consequences motivate adaption and mitigation efforts such as early warning
systems, protective infrastructure (e.g. storage, defenses, drainage, infiltration), public awareness and education, and zoning
regulations (Tumbare, 2000; Tauhid and Zawani, 2018; Charlesworth and Warwick, 2011).

Due to these growing flood risks, early warning systems, or forecasting systems, can help understand future conditions
and provide intelligence to furnish adequate warnings to protect life, prevent damages, and enhance resilience (Strömberg,
2007; Cools et al., 2016; UNISDR, 2015; Baudoin et al., 2014; Golnaraghi, 2012; UNEP, 2012; Liu et al., 2018a; Schumann
et al., 2013). The early warning of flood disasters at national scales requires the use of continental-scale, forecast hydrology
models and modeling frameworks that span intranational political boundaries. The applications of these models extend be-
yond early warning systems to provide historical trends for applications in infrastructure planning, public planning, insurance
underwriting, and more. The Office of Water Prediction (OWP), an office of the National Oceanic and Atmospheric Administra-
tion (NOAA) along with partners at the National Center for Atmospheric Research (NCAR), developed such a continental-scale
model known as the United States (US) National Water Model (NWM) (Salas et al., 2018; Gochis et al., 2021; Cosgrove et al.,
2019; Cohen et al., 2018; Oceanic and Administration), 2016; of Water Prediction, 2022). The NWM is based on a configura-
tion of the Weather Research and Forecasting Hydro (WRF-Hydro) model that accounts for land surface processes as well as
overland and channel routing (Gochis et al., 2021; Salas et al., 2018; Cosgrove et al., 2019). Operationally, the NWM produces
streamflow analysis and forecasts at multiple time horizons depending on location which include the conterminous United
States (CONUS), Puerto Rico, Hawaii, and portions of Alaska (Cosgrove et al., 2019; Oceanic and Administration), 2016;
of Water Prediction, 2022). The NWM routes streamflow across the NWM Version 2.1 (V2.1) stream network, based on the
National Hydrography Dataset Plus Version 2 (NHDPlusV2) network, is comprised of more than 5.5 million kilometers (kms)
of lines discretized into more than 2.8 million forecast points (Aristizabal et al., 2022). The NWM V2.1 stream network be-
longs to the NWM "hydrofabric" defined as a catalog of geospatial layers relevant to hydrology modeling including stream
network flowpaths, catchments, reservoirs, and more (of Water Prediction, 2022; Cosgrove et al., 2019). While streamflow is
an important variable for engineering and scientific applications of fluvial flooding, flood inundation stages, extents and depths
are much more tangible variables to the stakeholders flood events directly impact.

The shallow water equations, a system of two hyperbolic partial differential equations, formally govern the flow of fluvial
surface water by conserving both mass (first equation) and momentum (second equation) and can be expressed in both the



1-Dimensional (1D) (Saint Venant Equations) and 2-Dimensional (2D) forms. Solving this system in full 2D form requires numerical methods that can be very cost prohibitive and numerically unstable in an operational setting across continental-scales at high spatial discretizations (10 meter (m) or higher). This use case motivates the implementation of an inundation proxy, also known as a zero-physics or a simplified conceptual model, that is agnostic to the shallow water equations while still computing accurate fluvial inundation extents and depths (Teng et al., 2015; Bates and De Roo, 2000). Height Above Nearest

Drainage (HAND) detrends elevations within digital elevation models (DEMs) to compute drainage potentials by normalizing elevations to the nearest, relevant flowpath instead of datums that represent mean sea level (Rennó et al., 2008; Nobre et al., 2011, 2016). HAND as a terrain index has been used extensively for producing flood inundation maps (FIMs) from both modeled or observed stream flows and stages (Nobre et al., 2016; Afshari et al., 2018; Garousi-Nejad et al., 2019; Johnson et al., 2019; Zheng et al., 2018a, b; Zhang et al., 2018; Teng et al., 2015; Li et al., 2022b), as well as for assisting the remote

sensing detection of fluvial inundation (Aristizabal et al., 2020; Shastry et al., 2019; Aristizabal and Judge, 2021; Huang et al., 2017; Twele et al., 2016). HAND operates as an inundation proxy by thresholding the relative elevation (or HAND) values with a singular river stage value for each catchment corresponding to the drainage area of a given river reach (Nobre et al., 2016; Garousi-Nejad et al., 2019; Johnson et al., 2019; Zheng et al., 2018a; Teng et al., 2015; Liu et al., 2016; Maidment, 2017; Liu et al., 2018b, 2020, 2018a). When used to generate inundation extents and depths from streamflow, reach-averaged

synthetic rating curve (SRC) sample geometric variables along an entire reach and normalize using the length of the reach to create stage-discharge relationships (Zheng et al., 2018b; Aristizabal et al., 2022; Godbout et al., 2019). These relationships depend on the friction parameter, Manning's n, and are used to convert streamflows to stages for eventual 2D mapping with HAND. Numerous investigations have validated the use of HAND for flood mapping applications as a suitable alternative to more sophisticated physics-based techniques for large scale and high resolution use cases (Johnson et al., 2019; Li et al.,

2022a; Aristizabal et al., 2022; Nobre et al., 2016; Godbout et al., 2019; Afshari et al., 2018; Zhang et al., 2018; Teng et al., 2015, 2017; Diehl et al., 2021; Hocini et al., 2021; Bates et al., 2003).

Several prior and active large-scale HAND implementations catered to operational early warning systems applications including the National Flood Interoperability Experiment (NFIE) (Maidment, 2017; Liu et al., 2016, 2018b), GeoFlood (Zheng et al., 2018a; Hocini et al., 2021; D'Angelo et al., 2022; Carruthers, 2021; Zheng et al., 2022), and PyGFT (Petrochenkov

and Viger, 2020; Verdin et al., 2016). The NFIE was a broad, inter-institutional, and pioneering effort to apply HAND to the initial versions of the NWM which leveraged 1/3 arc-second (10 m) seamless elevation data available at the time (Maidment, 2017; Liu et al., 2016, 2018b) from the United States Geological Survey (US Geological Survey)'s National Elevation Dataset (NED) (Gesch et al., 2002; Gesch and Maune, 2007). Zheng et al. (2018a) applied HAND for operational applications with 1/27 arc-second (1 m) elevation data with a novel least cost, geodesic based stream delineation method (Passalacqua

et al., 2010, 2012; Zheng et al., 2018a, 2019; Carruthers, 2021; D'Angelo et al., 2022; Zheng et al., 2022). For applications with the NWM, an advanced version of HAND coupled with the use of SRCs, known as OWP FIM, converts NWM analysis, reanalysis, and forecast streamflows to river stages and fluvial inundation depths and extents on an operational basis to CONUS while extending the modeling domain to Puerto Rico and Hawaii (Aristizabal et al., 2022, 2023b). OWP FIM utilizes some of the latest datasets including the National Hydrography Dataset Plus High Resolution (NHDPlusHR) (Moore et al., 2019), Na-



tional Levee Database (NLD) (Engineerings, 2021), and the NWM V2.1 hydrofabric (of Water Prediction, 2022; Oceanic and Administration), 2016; nwm, 2021; Gochis et al., 2021). These datasets enforce hydrologically relevant features such as levees and the general location of flowpaths to facilitate conflation with the forecast stream network (Aristizabal et al., 2022, 2023b). Additionally, OWP FIM advanced a fundamental limitation of HAND that limits sourcing fluvial inundation only from the nearest, relevant flowpath (McGehee et al., 2016; Aristizabal et al., 2022; Zhang et al., 2018; Li et al., 2022a; Zheng et al.,

2018a, b; Nobre et al., 2016). Flowpaths of higher Horton-Strahler stream order that could contribute inundation to a given area have no way of extending beyond catchment lines which creates artificial bottlenecks in inundation extents, especially along junctions of high order rivers with their lower flow tributaries (Aristizabal et al., 2022; McGehee et al., 2016). To resolve this limitation, OWP FIM disaggregates the NWM V2.1 stream network into segments of effective unit stream order called level paths in a version of HAND called Generalized Mainstems (GMS) (Aristizabal et al., 2022). In terms of terrain data,

OWP FIM uses the 10 m DEM from the NHDPlusHR elevation dataset which is the elevation basis, derived in batches from 3-Dimensional Elevation Program (3DEP), for additional hydrography products within the NHDPlusHR (Aristizabal et al., 2022; Moore et al., 2019). The previous advances in OWP FIM stopped short of accounting for Light Detection and Ranging (LiDAR) point elevation observations (Aristizabal et al., 2022) that are now nearing their first collection cycle to form a novel seamless, continental scale DEM from 3DEP (USGS, 2021, 2022).

Broad scale terrain information in the form of DEMs is fundamental to all FIM models and a significant influence on inundation skill (Bales and Wagner, 2009; Dobbs, 2010; Wang and Zheng, 2005; Merwade et al., 2008; Witt III, 2015; Garousi-Nejad et al., 2019; Li et al., 2022b; Neal et al., 2011). The National Geospatial Program, under the US Geological Survey, is the primary authority on collecting, processing, and maintaining terrestrial elevation data within the US in collaboration with Federal partners within the National Digital Elevation Program (NDEP) (omb, 2016; Dewberry, 2011; Council et al.,

2007, 2009; Sugarbaker et al., 2014). The NED (Gesch et al., 2002; Gesch and Maune, 2007), forms the seamless elevation layers of the The National Map (TNM) (Gesch et al., 2009; Archuleta et al., 2017; Arundel et al., 2015a, 2018; Kelmelis et al., 2003). Prior to the introduction of 3DEP, TNM was originally composed of three seamless DEMs at 1/3 (10 m), 1 (30 m), and 2 (90 m) arc-second resolutions produced from a variety legacy sources including digital photogrammetry, cartographic contours, mapped hydrography, and elevations from Shuttle Radar Topography Mission (SRTM) (Gesch et al., 2002; Gesch

and Maune, 2007; Arundel et al., 2015a). High quality elevations derived from LiDAR and Interferometric Synthetic Aperture Radar (InSAR) have been integrated into TNM seamless elevation products as made available prior to and after the introduction of 3DEP (Snyder et al., 2013; Gesch et al., 2002; Arundel et al., 2015a). Work by Gesch et al. (2014),(Gesch and Maune, 2007), and Dobbs (2010) illustrated that the inclusion of higher quality elevation data sources had significant improvement in the accuracy of NED data when compared to the National Geodetic Survey (NGS) (Roman et al., 2010). Gesch et al. (2014)

identified that the NED 1/3 arc-second DEM, as of April 2013, had a mean error of -0.29 m with an root mean squared error (RMSE) of 1.55 m when compared to over 25 thousand reference points. At the time of evaluation, the NED was subject to legacy, lower quality data sources dating almost a century in the past (Sugarbaker et al., 2014; Gesch et al., 2014; Gesch and Maune, 2007). This reduction in error and its impact on people and commerce (Dewberry, 2011) motivated action on collection of elevation data from higher quality data sources (Sugarbaker et al., 2014).



3DEP is a national, multi-organizational effort by the NDEP to survey elevations with high quality sensors in response to growing stakeholder needs on a recurring collection cycle of no greater than 8 yrs (years) (Dewberry, 2011; Snyder et al., 2013; Sugarbaker et al., 2014). 3DEP leverages two main collection technologies including LiDAR for the CONUS, Hawaii, and US territories as well as InSAR for Alaska. LiDAR, the collection source of focus in this study, is a light emitting, reflection, and collection technology that beams concentrated powerful light of wavelengths between 1000 - 1600 nanometer (nm) (Muhadi

et al., 2020). The reflection of the light is collected while recording the travel time and intensity of return. LiDAR sensors are mounted on top of a variety of mobile or static platforms whose positions are geo-tracked as they collect LiDAR returns (Passalacqua et al., 2015). The travel time of the returns, along with knowledge of the speed of light, serve as a relative positioning of the target(s) referenced to a common vertical datum while the intensities serve as indicators of what the target(s) represent. Modes within the relationship of return intensities with respect to travel time/distance from the LiDAR wave forms

can be indicative of vegetation or other land use/land covers (LULCs) that reflect signals at varying distances and magnitudes and influence elevation errors (Gesch et al., 2014). These modes can be discretized into varying DEM products representing bare earth, structures, or canopy elevations. The horizontal and vertical accuracies and the horizontal resolutions of terrain observations derived from LiDAR, and even the consequential economic benefits (Dewberry, 2011, 2022), are dependent on a variety of sensor, platform, target, and collection specifications and practices such as nominal pulse spacing, nominal pulse

density, and LULC of the target (Heidemann, 2018; Passalacqua et al., 2015; Smith et al., 2019; Salach et al., 2018; Gesch et al., 2014). LiDAR produces point cloud datasets which are scattered, geo-referenced points representing full wave forms or discretized return intensities. Various assessments of the vertical accuracies of LiDAR point clouds have yielded satisfactory results in agreement with 3DEP requirements (Stoker and Miller, 2022; Kim et al., 2022; Callahan and Berber, 2022; Kim et al., 2022; Salach et al., 2018; Passalacqua et al., 2015). Point clouds must undergo a series of operations to produce analysis

ready, seamless DEMs (Passalacqua et al., 2015).

    3DEP extends TNM to include a 1/27 arc-second (1 m), LiDAR derived DEM product for CONUS, Hawaii, and US territories as well as a 1/2 arc-second (5 m) DEM for Alaska derived from InSAR (Sugarbaker et al., 2014; Stoker et al., 2015). To create bare earth DEMs, LiDAR observations must undergo a series of processes that filter out returns from vegetation, anthropogenic, and other features then grid the observations with resampling methods (Passalacqua et al., 2015). The 1 m 3DEP

product is a hydrologically conditioned (hydro-flattened), topographic, and bare-earth raster DEM gridded to 1 km square shaped tiles with 6 pixels of overlap (Arundel et al., 2015b). Hydro-flattening refers to a process in which hydrologic features such as lakes, reservoirs, streams, rivers, and more are flattened in elevation for bathymetric regions from lower bank to lower bank represented by breaklines (Archuleta et al., 2017; Maune and Nayegandhi, 2018). This flattening excludes along gradient directions, parallel to the direction of the breaklines, for hydrologic features that naturally exhibit water conveyance such as

streams, rivers, and long reservoirs (Arundel et al., 2015b). This process includes elevations underneath bridges that are not accurately observed from topographic LiDAR (Bales and Wagner, 2009). According to specifications, the horizontal accuracy of 1 m 3DEP is within 1 m while the vertical accuracies are within 19.6 centimeters (cms) and 30 cm at the 95% confidence interval for non-vegetative and vegetative regions, respectively (Arundel et al., 2015b; Heidemann, 2018). Non-vegetative vertical accuracies fall within 10 cm RMSE (Arundel et al., 2015b; Heidemann, 2018). Work by Stoker and Miller (2022),



Callahan and Berber (2022), and Kim et al. (2022) have verified the vertical accuracies and general quality of the DEMs for
3DEP specifications.

The quality of FIM extents are subject to a wide variety of terrain related factors including collection technology, gridding
methods, resampling techniques, hydrological conditioning processes, presence of bathymetry, vertical accuracies and hori-
zontal resolutions (Merwade et al., 2008). The main enhancement of including 3DEP data within HAND based OWP FIM is
the broader availability of high quality data sources for elevations such as LiDAR with enhanced vertical accuracies and hor-
izontal resolutions (Arundel et al., 2015b; Stoker and Miller, 2022; Archuleta et al., 2017). Generally speaking, the literature
has demonstrated the sensitivity to and improved effect of using 3DEP or LiDAR data in general has on the quality of FIM
extents mostly due to the (Podhorányi and Fedorcak, 2015; Bales and Wagner, 2009; Merwade et al., 2008; Witt III, 2015; Ma-
son et al., 2007; Zheng et al., 2018a). Limitations have been noted with respect to vertical accuracies in areas with vegetation,
buildings, bridges, or classified as bathymetric (Merwade et al., 2008; Mason et al., 2007; Bales and Wagner, 2009; Podhorányi
and Fedorcak, 2015). FIM extents in areas of low topographic relief or areas behind natural or anthropogenic flow divides can
be very sensitive to vertical accuracies (Sanyal and Lu, 2004; Garousi-Nejad et al., 2019; Godbout et al., 2019; Jafarzadegan
and Merwade, 2017; Papaioannou et al., 2017). Specifically for HAND, research by Zheng et al. (2018a) and Garousi-Nejad
et al. (2019) noted improvement when utilizing higher resolution LiDAR derived DEMs for HAND based FIM.

The spatial resolution of topography likely interacts with many other sources of FIM extent uncertainties including but not
limited to elevation source quality, LULC, streamflow intensities, physics employed, and model parameterizations (Fewtrell et
al., 2008; Savage et al., 2016; Neal et al., 2011; Thomas Steven Savage et al., 2016). Numerous researchers have evaluated
resolution more generally across the spectrum of FIM models to focus more on urban areas where resolution could play an
integral part into determining extents (Fewtrell et al., 2008; Neal et al., 2011; Ozdemir et al., 2013; Muthusamy et al., 2021;
Savage et al., 2016; de Almeida et al., 2018; Dixon and Earls, 2009). While the studies evaluating HAND are extensive
(Afshari et al., 2018; Nobre et al., 2011; Garousi-Nejad et al., 2019; Godbout et al., 2019; Speckhann et al., 2018; McGrath
et al., 2018; McGehee et al., 2016; Li et al., 2020, 2022b, a; Liu et al., 2016, 2018b, a; Li and Demir, 2022; Liu et al., 2020;
Aristizabal et al., 2022; Maidment, 2017; Zheng et al., 2018a, b, 2019, ?; Diehl et al., 2021; Johnson et al., 2019; Jafarzadegan
and Merwade, 2019), only a few studies have investigated the effects of high quality DEMs and their spatial resolutions on
FIM extents when derived from HAND. Li et al. (2022b) evaluated HAND based FIM over a small domain and concluded
that resampled LiDAR performed best at the 5 m spatial resolution when compared to coarser, resampled grids. Zheng et al.
(2018a) incorporated LiDAR derived elevations while also incorporating a novel stream delineation method and concluded
that both combined performed better than utilizing legacy NED 10 m datasets with NHDPlusV2 hydrography as the datum
for HAND computation. Lastly, working in flat areas with some anthropogenic influence, Garousi-Nejad et al. (2019), used a
3 m DEM and found improvement in FIM quality extents when compared to the use of a 10 m DEM derived from different
sources. In contrast to the other studies, Speckhann et al. (2018) evaluated the sensitivity of HAND based FIM extents to DEM
resolution in Brazil using DEMs from SRTM and found little to no effect given this region. Both Garousi-Nejad et al. (2019)
and Zheng et al. (2018a) highlighted the importance of high resolution elevations and novel stream delineation tools to avoid
the negative effects of little to no bathymetric information. Due to the interacting uncertainties and the dearth of research on





this question with respect to HAND, it is difficult to conclude what the effect would be on the quality of HAND based OWP
FIM by incorporating the latest 3DEP data at varying spatial resolutions.

As the spatial coverage of the 3DEP 1 m product rapidly approaches CONUS scale in 2023, we investigate the integration
of 3DEP data into OWP FIM for model specific evaluation (USGS, 2021, 2022). We use 3DEP data for the HAND compu-
tation process to generate the FIM hydrofabric. OWP FIM uses novel combination of input datasets, hydrological condition-
ing (hydro-conditioning) processes, level path scale processing, and parameterizations to produce HAND and these specific
combinations of methods could interact with terrain related variables including source and resolution. Additionally, we investi-
gated the utility of varying spatial resolutions from 3, 5, 10, 15, and 20 m, specifically its effect on FIM extents. HAND depends
on the drainage assumptions which requires DEMs to undergo a long series of enforcement processes to ensure monotonically
decreasing elevations with hydrologically correct flow directions (Garousi-Nejad et al., 2019; Nobre et al., 2011, 2016; Aris-
tizabal et al., 2022). The resampling of DEMs into varying spatial resolutions could interact with these hydro-conditioning
operations thus influencing the FIM hydrofabric and the resulting quality of the FIMs produced. OWP FIM is scheduled for
public release in 2023 for a region covering 10% of the US population. Evaluations are needed specifically for this region
and how 3DEP elevations at varying resolutions affect skill. As validation, we used 1D Hydrologic Engineering Center River
Analysis Center (HEC-RAS) modeled flood inundation extents from both the Base Level Engineering (BLE) published by the
Interagency Flood Risk Management (InFRM) team. By varying the spatial resolution of 3DEP DEMs, we seek to quantify the
relationship in an empirical fashion between spatial resolution and FIM skill produced from HAND that requires significant
DEM manipulations to satisfy inherent assumptions. For analysis purposes, we consider a series of potential, catchment scale
explanatory variables with multi-variable regression analysis to help explain some the of the catchment scale variation in the
metrics we employed that describe agreement with the BLE FIMs.

## 2   Materials and Methods

### 2.1   Overview

Investigating the effects of LiDAR derived DEMs and their spatial resolutions involved a multi-step process of data curation,
production, evaluation, and analysis. Source information was gathered to produce HAND and its associated datasets most
specifically the DEMs from multiple sources and spatial resolutions including 3, 5, 10, 15, and 20 m. The FIM hydrofabric,
or the collection of datasets required to convert streamflows to FIM extents, was produced using these various DEMs (Aristi-
zabal et al., 2022; Aristizabal and Judge, 2021). FIMs were produced by intersecting the BLE cross sections, as described by
Aristizabal et al. (2022), furnished by the InFRM team for both the 1% (100 year (yr)) and 0.2% (500 yr) recurrence flows
(fem, 2016, 2021a, b; sta, 2019a, b, c, d, e, f, g). These intersected flows were converted to reach-averaged stages using SRCs
(Aristizabal et al., 2022; Liu et al., 2016, 2018b; Zheng et al., 2018b). Reach-averaged stages were used to threshold HAND
values on a per catchment basis which translates to a flooded pixel when the stage value exceeds zero (Zheng et al., 2018b;
Aristizabal et al., 2022). Catchments are defined here as the unique surface drainage areas assigned to each river reach. These
extents at the 100 and 500 yr flow magnitudes were then compared to the original BLE furnished extents for the corresponding





magnitudes. Agreement statistics and maps were computed for binary categorical variables (inundated = positive and not inundated = negative) then resampled to the catchment scale. A number of covariates and factors were selected at the catchment

scale for analysis purposes to explain some of the catchment to catchment variance in the selected metrics with the help of regression models. A high level graphical summary of this explanation is furnished in Figure 1.

## 2.2 Datasets

We used a wide variety of datasets for investigating the effects of DEM source and spatial resolution on FIMs produced from HAND. We compared legacy DEMs, used by Aristizabal et al. (2022), sourced from the NHDPlusHR program that were

available for the entire NWM domain at the Hydrologic Unit Code (HUC)-4 scale. As the 3DEP program rapidly approaches continental scale availability, the inclusion of 3DEP DEMs were considered here at various spatial resolutions (USGS, 2022; Stoker et al., 2015; USGS, 2021; Chegini et al., 2021; Survey, 2022). Aristizabal et al. (2022) provided more details for the datasets listed in Figure 1 denoted by "Other Datasets". The remaining datasets used in this study for DEM experimentation and for analysis are elaborated on in Table 1. The analysis datasets are those used to help explain some of the spatial variation in

the metrics. NWM catchments were used to resample the agreement maps and metrics down to the catchment scale. Analyses of the importance of various attributes within NWM catchments considered flowpath properties such as channel slope, length, presence of reservoirs, and catchment area. Other attributes, aggregated to the catchment scale for consideration, included terrain slope, imperviousness, overland roughness, and LULC from the National Land Cover Database (NLCD) which were used as either covariates or factors in the statistical analysis of this study.

## 2.3 Data Retrieval

We used Hydroclimate Data Retriever (HyRiver) (Chegini et al., 2021) for retrieving topographic, land use/land cover, imperviousness, and overland roughness data. HyRiver is a suite of nine open-source Python packages that provide access to a wide variety of hydrology and climatology datasets within the US, through web services. In this study, we used two of these packages: Py3DEP and Python Hydrogeological Datasets (PyGeoHydro).

Py3DEP provides access to 3DEP's dynamic and static services. The dynamic service retrieves topographic data at any resolution using the best available raw elevation data for a requested region, whereas the static service only provides DEM data at 10, 30, and 60 m resolutions. It has some other utilities, including querying availability of raw elevation data at different resolutions from various sources. In this study, we used the dynamic service to obtain DEM at different resolutions, and the raw data availability functionality of Py3DEP for determining the highest resolution data available in the region of our study.

While Py3DEP is developed only for retrieving topographic data from a single source, PyGeoHydro can query various types of data from different sources, e.g., National Inventory of Dams (of Dams , NID), Watershed Boundary Dataset (, WBD), and National Land Cover Database (, MRLC). It also includes additional functionalities, including a look-up table for associating overland roughness to land cover type based on Liu et al. (2019). In this study, we obtained HUC geometries, reservoirs, LULC, imperviousness, and overland roughness data using PyGeoHydro.



**Table 1.** Dataset sources, names, descriptions, and citations. Datasets used to generate HAND except for the DEMs are not listed in detail here but explained by Aristizabal et al. (2022).

| Source | Dataset | Description | Citations |
|---|---|---|---|
| USGS | 3DEP | Continental-scale high resolution (1 m) DEMs from high quality sources. | USGS (2022, 2021); Stoker et al. (2015); Chegini et al. (2021) |
| USGS | NHDPlusHR DEM | DEMs available from NHDPlusHR program at the HUC-4 level for the entire country. | nhd (2021); Moore et al. (2019) |
| Various | Other Datasets (See Figure 1.) | Other datasets used for production of HANDs based FIM hydrofabric. | Aristizabal et al. (2022) |
| InFRM | Flood Inundation Extents | Inundation depths produced by InFRM BLE HEC-RAS 1D for 1% and 0.2% recurrence interval events. | fem (2021b, 2016, 2021a); sta (2019a, b, c, d, e, f, g) |
| InFRM | Cross-Sections | BLE HEC-RAS 1D cross-sections for 100 yr (1%) and 500 yr (0.2%) streamflow magnitudes used to intersect with NWM reaches. | fem (2021b, 2016, 2021a); sta (2019a, b, c, d, e, f, g) |
| OWP | NWM Catchments | Surface drainage area corresponding to each reach in the NWM adapted from National Hydrography Dataset Plus (NHDPlus) V2 Catchment feature class. | nwm (2021) |
| OWP | NWM Stream Network | Stream network flowpaths used by NWM for routing and forecasting adapted from NHDPlus V2 NHDFlowline Network feature class. | nwm (2021) |
| USGS | Terrain Slope | Terrain slope (vertical/horizontal) computed from 3DEP DEMs. | Dewitz (2021); Yang et al. (2018); Chegini et al. (2021); Survey (2022) |
| NLCD | LULC | LULC as produced by the NLCD 2019 at the 30 m resolution derived partly from LandSat imagery. | Dewitz (2021); Yang et al. (2018); Chegini et al. (2021); (MRLC) |
| NLCD | Imperviousness | Urban impervious surface as a percentage of developed surface over 30 m pixels. | Dewitz (2021); Yang et al. (2018); Chegini et al. (2021); (MRLC) |
| NLCD | Overland Roughness | Overland Roughness or Manning's n for given pixel within NLCD. | Dewitz (2021); Yang et al. (2018); Chow (1959); Chegini et al. (2021); (MRLC); McCuen et al. (2005); Kalyanapu et al. (2009) |




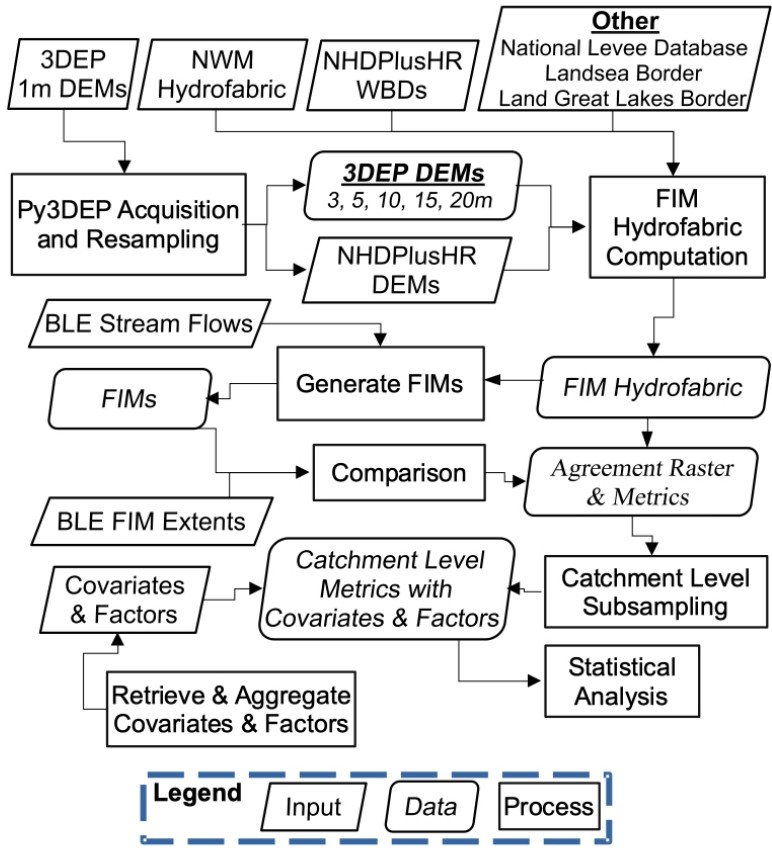

**Figure 1.** Figure illustrates the overall process for generating HAND and evaluating the use of LiDAR derived DEMs and their resolutions. Input datasets were collected from two different source DEMs, the 3DEP and NHDPlusHR. The 3DEP DEMs were resampled with Python 3-Dimensional Elevation Program (Py3DEP) to 3, 5, 10, 15, and 20 m spatial resolutions. Both source DEMs and their resolutions were used to compute the FIM hydrofabric which is comprised of various datasets used to produce FIM including HAND, catchments, and SRCs. BLE cross sections were intersected with the NWM stream network to obtain stream flow estimates. These estimates were used to produce FIM using the SRCs coupled with HAND to produce estimates of the 100 and 500 yr extents. These extents were then compared to the extents from the BLE thus removing hydrology related errors that could be introduced if NWM streamflows were used. The agreement statistics were resampled to the NWM catchment level then referenced to a long series of catchment level covariates and factors that were used for statistical analysis and inference.



## 2.4 DEM Preparation

DEMs underwent a curation procedure prior for use with HAND computation and our experimental design. To match the existing framework with the use of NHDPlusHR DEMs, Py3DEP was used to query the image server to acquire 3DEP elevations at a HUC-4 scale. To counter pixel limitations within the web service, queries were completed using overlapping tiles and mosaiced together using virtual rasters (VRTs) (Survey, 2022). To investigate the effect of varying spatial resolutions on FIM skill and computational performance, queries were elected taken at 3, 5, 10, 15, and 20 m spatial resolutions. Utilizing the "check_3dep_availability" tool, we determined that 1 m 3DEP information is available for the entire study region (see Section 2.5). Py3DEP queries a US Geological Survey dynamic web service for the best available DEM when generating its mosaics and resamples them given the user furnished resolution (Survey, 2022). Use of the Py3DEP function "query_3dep_source" confirmed availability of 1 m LiDAR data for the entire study area which means the functionality uses it for resampling purposes (Chegini et al., 2021; Stoker et al., 2015; Survey, 2022; USGS, 2022, 2021). The availability of the 1 m data as well as the other available source DEMs are illustrated in Figure 2. The selected resolutions of 3, 5, 10, 15, and 20 m were bounded by computational demands since further optimizations within the code and external dependencies should be considered prior to transitioning to 1 m elevation information for HAND computation.

## 2.5 Study Area

The site selection process considered several factors. The location of the site was limited by the availability of validation (discussed in Section 2.6) as well as the availability of 1 m 3DEP information (USGS, 2022, 2021). Additionally, the location of the site was influenced by OWPs plan to release FIM services in stages as a function of the percentage of the population served. The first release will serve 10% of the US population and cover portions of East Texas (TX) as well as the Mid-Atlantic states. On the other hand, the size of the evaluation site was constrained by the computational burdens of producing the FIM hydrofabric at multiple resolutions.

With these criteria in mind, we selected the Neches River sub-region as the study area for this experiment. The HUC-4 (1202) sub-region comprises 7 HUC-8 sub-basins, ranging continuously from 12020001 to 12020007. Located in South-East TX near the Louisiana border, the site stretches from Tyler to Beaumont, and includes the towns of Nacogdoches and Lufkin as depicted in Figure 3. Numerous braided streams and 15 reservoirs, including one of the largest ones, the Sam Rayburn Reservoir, populate the study area. Figure 4 depicts the spatial distribution of LULCs as defined in the 2019 NLCD. The study area features a low slope, with low-lying areas mostly comprising four LULCs: evergreen forests (31.1%), pasture/hay (17.2%), woody wetlands (16.7%), and mixed forests (11.4%). The developed LULCs together account for only 7.3% of the site's area. In summary, the study area has low terrain slope and minimal anthropogenic influence.

## 2.6 Evaluation

We chose the BLE FIM extents for evaluation, which are HEC-RAS 1D based models provided by InFRM and Federal Emergency Management Agency (FEMA) (fem, 2016, 2021a, b; sta, 2019a, b, c, d, e, f, g). FEMA's Region 6 publishes these FIMs,



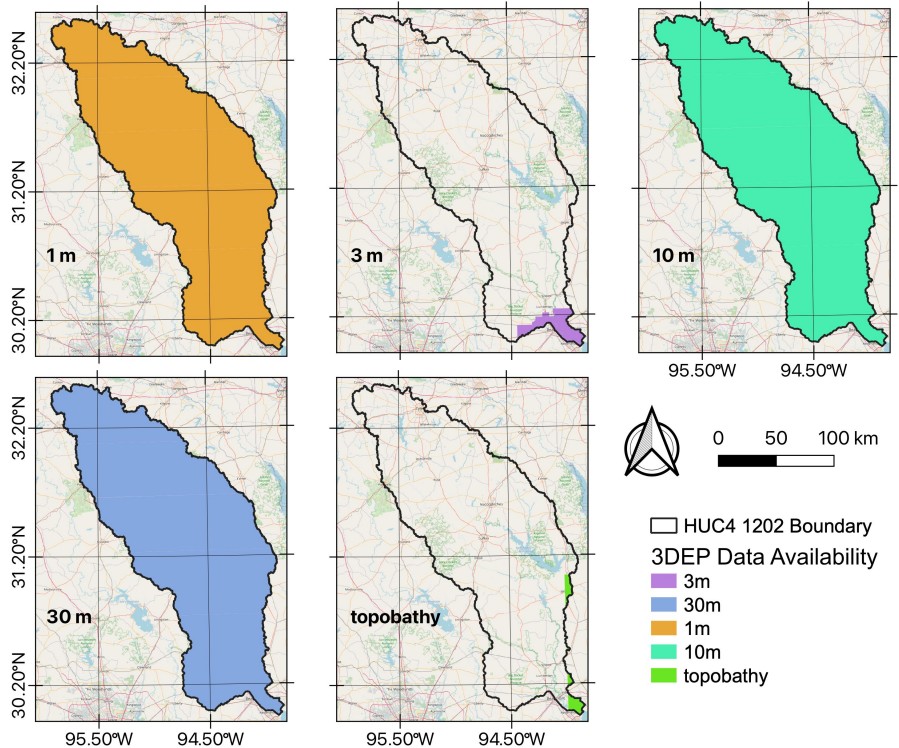

**Figure 2.** Illustrates the source DEMs available within 3DEP for the study area. High resolution 1 m information is available for the entire study area meaning it was used as the source resolution for resampling to the resolutions used for HAND computation including 3, 5, 10, 15, and 20 m. See Section 2.5 for more information within the study area. ©OpenStreetMap contributors 2023. Distributed under the Open Data Commons Open Database License (ODbL) v1.0

which are available at both 1% (100 yr) and 0.2% (500 yr) flow magnitudes, and they also include cross-sectional information with the associated flows for each level. Despite being a modeled data set, HEC-RAS appears frequently in literature for comparison purposes, as it is an engineering scale model (Cook and Merwade, 2009; Rajib et al., 2016; Zheng et al., 2018a; Afshari

et al., 2018; Wing et al., 2017; Criss and Nelson, 2022; Follum et al., 2017). We chose to intersect the cross-sections with NWM flowpaths to remove errors and uncertainties associated with hydrological and meteorological inputs used to produce streamflows within the NWM (Aristizabal et al., 2022). This process enabled us to associate BLE derived 100 and 500 yr streamflow magnitudes with NWM forecasting points. If multiple intersections occurred per NWM stream reach, we took the median flow value. Even though this process may lead to conflation errors, we believe it allows for a better comparison with BLE FIM

extents by removing any errors introduced from variances in other hydrological processes outside of inundation (Aristizabal et al., 2022). For more detailed information on this technique and its application, see the evaluation methods in Aristizabal et al. (2022). These BLE FIMs have a spatial resolution of 3 m, making them suitable for evaluating high-resolution data. This spatial resolution also prevented us from using the 3DEP at its native 1 m resolution, which we deemed not meaningful





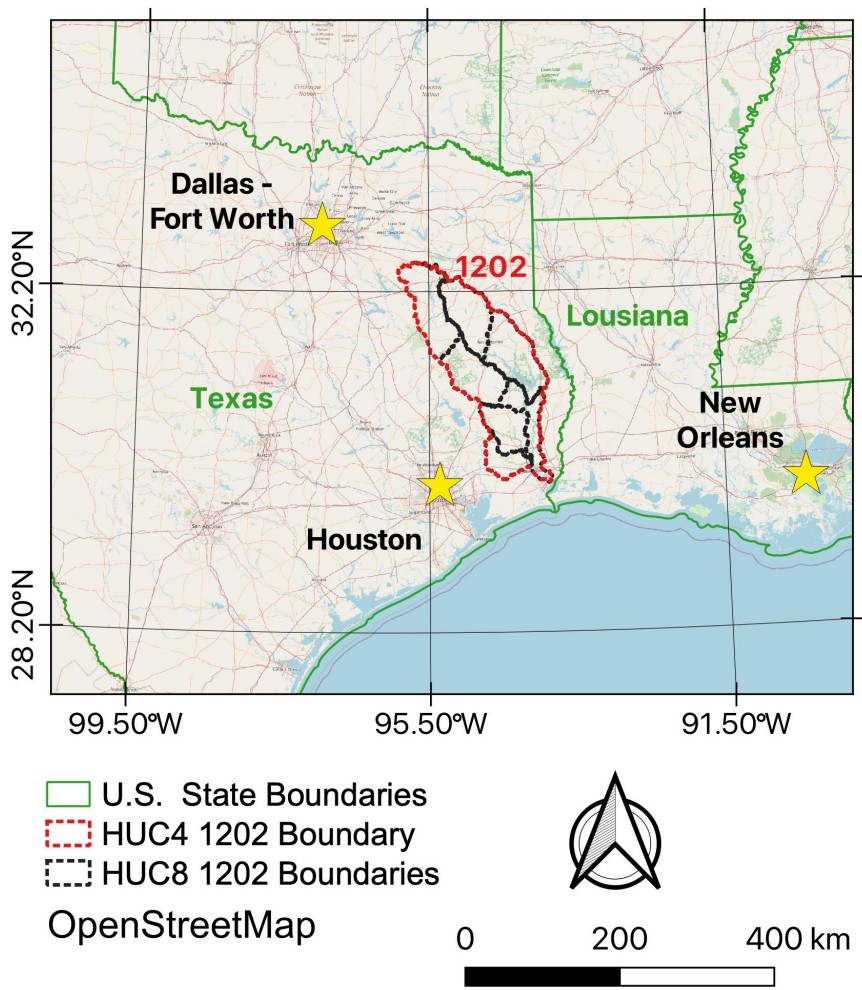

**Figure 3.** Overview of study area nestled in South East TX near the Louisiana border. Known as the Neches River sub-region or HUC-4 1202, the site is composed of 7 sub-basins or HUC-8s. ©OpenStreetMap contributors 2023. Distributed under the Open Data Commons Open Database License (ODbL) v1.0

for comparison. Moreover, producing HAND with 1 m information is computationally very expensive, leading to substantial
increases in central processing unit (CPU) time and memory usage, which we discuss in Sections 3 and 4.

In order to quantify agreement with the BLE FIM extents, we elected to apply binary contingency statistics. The primary metrics calculated within a contingency table include true positives (TPs), false positives (FPs), false negatives (FNs), and true negatives (TNs). We again note that the positive condition is considered inundated, while the negative condition is considered not inundated. In order to summarize the contingency table into secondary metrics, we employed the commonly used metrics





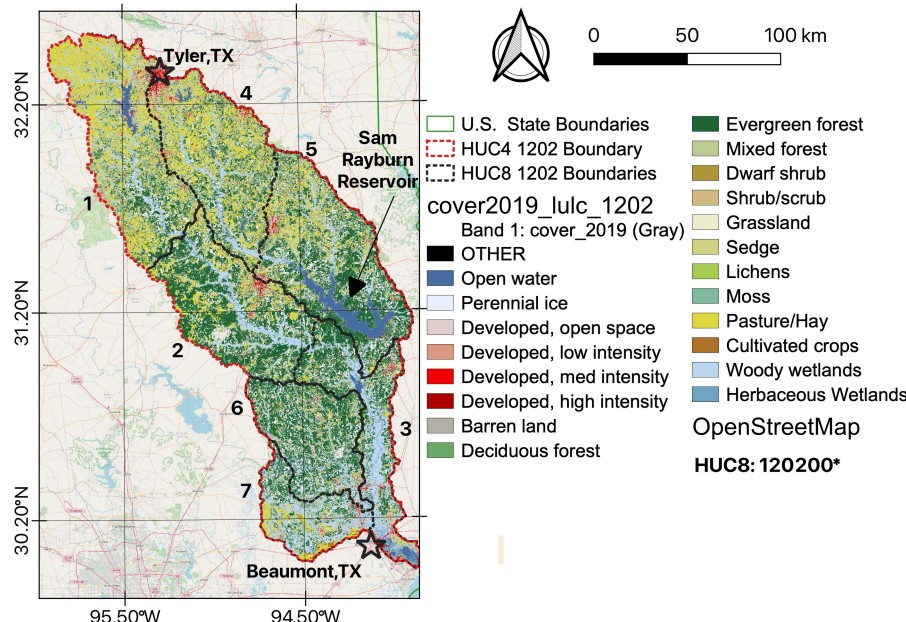

**Figure 4.** A detailed view showing the spatial distribution of the 2019 NLCD LULCs. About three-quarters of the site is made up of just four land covers including evergreen forest (31.1%), pasture/hay (17.2 %), woody wetlands (16.7 %), and mixed forests (11.4 %). Only about 7.2% of the site is considered developed. ©OpenStreetMap contributors 2023. Distributed under the Open Data Commons Open Database License (ODbL) v1.0

within flood modeling including critical success index (CSI), true positive rate (TPR), and false alarm rate (FAR) shown in Equations 1, 2, and 3, respectively (Gerapetritis and Pelissier, 2004; Schaefer, 1990).

$$CSI = \frac{TP}{TP + FP + TN} \tag{1}$$

$$TPR = \frac{TP}{TP + FN} \tag{2}$$

$$FAR = \frac{FP}{TP + FP} \tag{3}$$

TPR, also known as sensitivity, recall, probability of detection or hit rate, was used to describe a models ability to detect flooding as it represents performance in regions that are considered flooded within the benchmark. It is formally described as the proportion of inundated pixels that are accurately detected as flooded. FAR, also known as false discovery rate, the inverse of precision, or the inverse of positive predictive value, conveys the opposite since it is used to represent over-prediction. This





is formally described as the proportion of pixels incorrectly predicted as flooded with respect to the total number of pixels

predicted as flooded. Work in Gerapetritis and Pelissier (2004) illustrated how these two metrics are mathematically related
to CSI where only inundated regions in the benchmark dataset are ignored thus TNs are not considered. CSI is considered
unequitable or exhibit frequency dependency, which could limit its use in comparing predicted datasets in situations with
varying frequencies (Gerapetritis and Pelissier, 2004; Schaefer, 1990).

While these 3 widely adopted metrics are considered highly interpretable, we elected to include Matthew's Correlation Co-
efficient (MCC), shown in Equation 4, which is considered more equitable when dealing with cases of extreme class imbalance
(Chicco and Jurman, 2020; Chicco et al., 2021a, b; Boughorbel et al., 2017). However, it does value both conditions (inundated
and not inundated) to have equal impact (Chicco and Jurman, 2020; Chicco et al., 2021a, b; Boughorbel et al., 2017).

$$MCC = \frac{TP \cdot TN - FP \cdot FN}{\sqrt{(TP+FP)(TP+FN)(TN+FP)(TN+FN)}} \tag{4}$$

### 2.6.1 Analysis

Evaluations for this HUC-4 study region were conducted at the HUC-8 scale which produces seven HUC-8 metric values
across all five spatial resolutions evaluated as well as both flood magnitudes yielding about 70 samples to analyze ($7 \cdot 5 \cdot 2 =$
70). Analysis at this large HUC scale tends to erode away valuable information that could be used if a finer grain unit of
measurement were used instead. Under this justification, we opted to sub-sample agreement maps down to the NWM catchment
scale and recompute each of the four metrics for each catchment. There are 5,786 NWM catchments available for this study
area which generates 405,020 effective samples to analyze ($70 \cdot 5,786 = 405,020$). This yielded a much finer grain spatial
distribution of performance but also enabled the introduction of covariates and factors that can help explain some of the
catchment to catchment variance in the metrics. We define factors as categorical variables used as explanatory or predictor
variables to help form causal relationships that explain the catchment-level variances in the four metrics employed. The term
covariate serves the same function as the factors with the only distinction being that covariates are of continuous data types.
Many of these covariates and factors stemmed from NWM catchments or flowpaths themselves including channel slope,
catchment area, stream order, and reservoir.

The term reservoir here is used with respect to catchments that intersect NWM reservoirs. While NWM reservoirs are masked
out for evaluation and also not modeled for within OWP FIM, the BLE FIM extents do model reservoir inundation. This creates
regions of BLE inundation that extend beyond NWM reservoir definitions thus leading to FNs. NWM catchments that intersect
with NWM reservoirs are denoted as reservoir catchments and used as a factor to help account for the performance within these
regions. This is better illustrated in Figure 5.

In addition to catchment level attributes within the NWM hydrofabric, we collected a variety of datasets associated with
hydrological processes including NLCD LULC, imperviousness, overland roughness, and terrain slope. These factors and
covariates were obtained utilizing the HyRiver suite of tools described in Section 2.3. Overland roughness was determined by
the NLCD LULCs and previous research assigned coefficients for each category (Dewitz, 2021; Yang et al., 2018; Chow, 1959;
Chegini et al., 2021; , MRLC; McCuen et al., 2005; Kalyanapu et al., 2009). In order to aggregate to catchment scale, LULC was





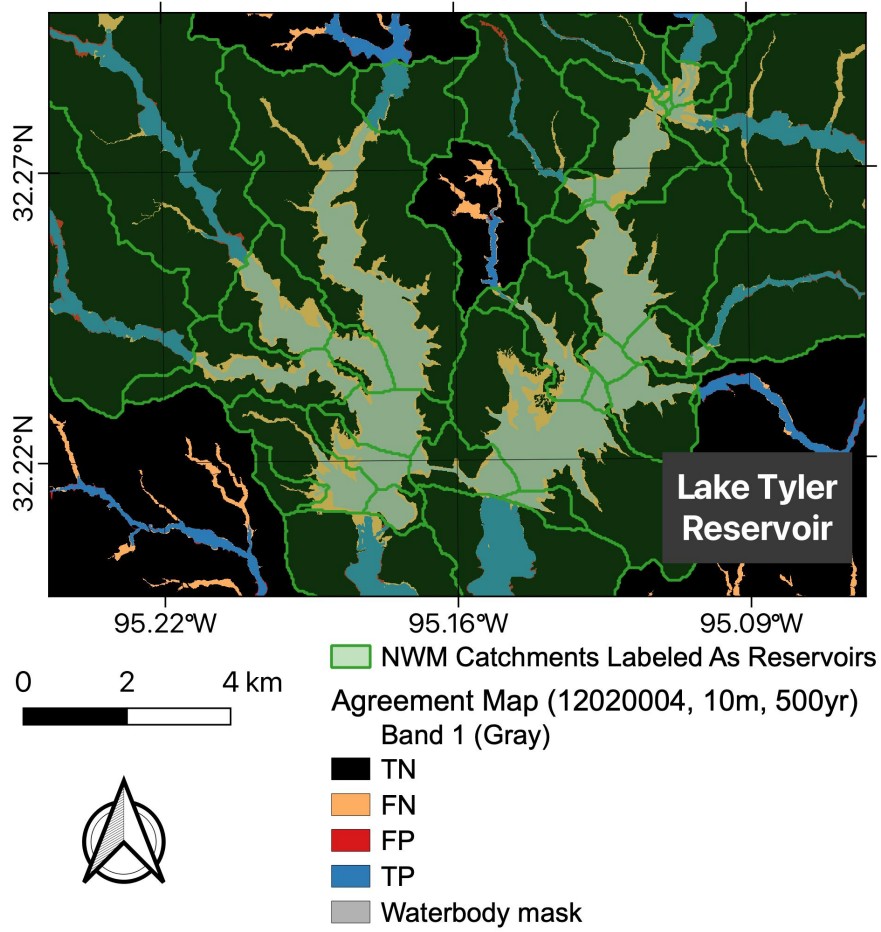

**Figure 5.** Figure shows Lake Tyler reservoir within HUC-8 12020004 of the study area. Background represents an agreement map between OWP FIM and BLE FIM at a 10 m spatial resolution for the 500 yr magnitude. Gray area shows masked out NWM reservoir since these are not being modeled within OWP FIM. The NWM catchments shaded in green represent catchments associated with this NWM reservoir as they are show to spatially intersect. These reservoir catchments were used in analysis to quantify the catchment variance in performance, partly, due to not accounting for reservoir inundation within BLE FIM.

taken as the dominant category by catchment while the covariates imperviousness, overland roughness, and terrain slope were aggregated by taking the catchment level mean value. This procedure created a total of 10 catchment level covariates and factors summarized as spatial resolution, DEM source, channel slope, catchment area, stream order, reservoir, LULC, imperviousness, overland roughness, and terrain slope. These covariates and factors are collectively known as features, predictors, explanatory variables, or independent variables, and were used to correlate to dependent, response, or outcome variables which were the four metrics of interest in this study. These are described in more detail in Table 2.




**Table 2.** Summary of catchment level covariates and factors used for statistical analysis. Includes the dataset, its statistical data type for analysis (factor or covariate), its units if it's a covariate or its levels if it's a factor.

| Dataset | Statistical Data Type | Levels for factors or Units for Covariates) |
| --- | --- | --- |
| Spatial Resolution | Factor | Five levels in meter units: 3, 5, 10, 15, & 20. |
| DEM source | Factor | Two levels: NHDPlusHR & 3DEP. |
| Channel Slope | Covariate | Vertical/horizontal as percentage. |
| Catchment Area | Covariate | Surface area in $km^2$. |
| Horton Strahler Stream Order | Factor | Six levels: 1 - 6 |
| Reservoir | Factor | Two levels: True (1) and False (0). |
| NLCD 2019 Dominant LULC | Factor | Fifteen levels: Woody Wetlands, Pasture/Hay, Evergreen Forest, Developed, Low Intensity, Shrub-Forest, Mixed Forest, Developed, Open Space, Cultivated Crops, Developed, Medium Intensity, Emergent Herbaceous Wetlands, Water, Herbaceous-Forest, Developed, High Intensity, Deciduous Forest, and Grasslands/Herbaceous. |
| Imperviousness | Covariate | Percent of pixel area that is impervious surface. |
| Overland Roughness | Covariate | Unitless: Friction coefficient for overland water flow. |
| Terrain Slope | Covariate | Vertical/horizontal as percentage. |
| Stream Order | Covariate | Horton-Strahler stream order as defined in NWM stream network. |

Given the fact that we aggregated a variety of catchment scale features for each associated catchment scale metric, we used the regression analysis to help explain the magnitude and significance of the linear relationships between the explanatory variables and the four responses (metrics: MCC, CSI, TPR, and FAR) (Montgomery et al., 2021; Chatterjee and Simonoff, 2013; Merrill et al., 2017). We avoided including the metrics with the NHDPlusHR DEM in the regression analysis since it was already clear that using 3DEP DEMs led to significant skill improvements. To build our regression model, we opted to use forward model selection of all one-way and two-way interactions utilizing the Akaike Information Criterion (AIC) and terminating the model selection after a minimum is reached. Explanatory variables were feature scaled from 0 to 1 prior to fitting to better compare across explanatory variables. Meaning, this procedure added variables to the regression model for each metric first. Finding the explanatory variable that minimized AIC, it left that variable in the model then moved on the remaining variables as long as the previous variable explained a minimum of 0.001 more than the previous model. This process helps build models with explanatory power while avoiding unnecessary complexity.

## 3 Results

Based on the observation of our results, we took an in-depth analysis of the effects of utilizing 3DEP DEMs first when compared to the legacy, NHDPlusHR DEMs. After confirming the positive effect of using 3DEP information, we varied the





spatial resolution of these DEMs and observed the impact on performance. To further investigate the effects of additional explanatory variables, we used investigate multiple linear model building with forward model selection to help explain some of the catchment to catchment variance in the four metrics. Lastly, we decided to do an in-depth analysis on a few of these

variables that we found of importance.

## 3.1 3DEP Data

For the given study area, we decided to investigate the effect on HAND based FIMs extents by utilizing the 3DEP data instead of the legacy source DEMs from the NHDPlusHR. We conducted this comparison on a NWM catchment scale in order to have a sense of the distribution of the results across some spatial definition finer than the HUC scale. Additionally, this comparison

was conducted by resampling the 3DEP 1 m data to a spatial resolution of 10 m to match that of the legacy DEM. Figure 6 details the results of this comparison in a scatter plot format. Each individual data point represents a sample of the metrics taken at the NWM catchment scale. The points are sampled across two axes representing their performance with NHDPlusHR DEMs on the x-axis and 3DEP DEMs on the y-axis. The 45 degree, diagonal line represents a dividing line where the metric values for both DEMs are the same. Catchment samples symbolized in green represent enhanced FIM extents for that catchment for

the given case while sample symbolized in red signify poorer quality extents. We also included descriptive statistics on each sub-figure representing the mean and standard deviation of the metric differences across DEMs (3DEP - NHDPlusHR) as well as the percentage of differences greater than or less than zero.

Overall, the use of the higher quality, more recently produced 3DEP DEMs generally enhances FIM extents across all the metrics and magnitudes examined. This is evident by observing the high proportion of catchments represented in green as well

as the high percentage of samples greater than zero for the first three metrics. The FAR is minimized so a lower proportion of samples above zero is considered better. Overall, approximately four in every five catchments in considered to benefit from the use of 3DEP when compared to the use of NHDPlusHR. This approximate relationship holds true across metrics and event magnitudes for our given experimental design.

## 3.2 Regression Analysis

After we established the effect of the new elevation data source on FIM extents, we elected to conduct regression analysis on the remaining explanatory variables of interest. As explained in the methods, we regressed on the four metrics of interest independently and fit the model in a forward selection fashion utilizing AIC as a measure of model fit. Figure 7 represents the resulting models from that forward model selection in graphical form. The four subplots represent the results of the model fit to each metric or response variable. The y-axis labels represent explanatory variables starting with the intercept followed

by the remaining variables and their two-way interactions in the order of selection as per the AIC metric. The points on the graph represent the values of the coefficients while the shape represents the level of significance from $>= 0.05$ (circle), $< 0.05$ (pentagon), $< 0.01$ (triangle), and $< 0.001$ (star). The green and red colors represent the nature of the effects as either positive (direct) or negative (indirect), respectively. Lastly, since AIC lacks interpretability, we elected to show the coefficient of determination or $R^2$ at each step of the forward selection process.



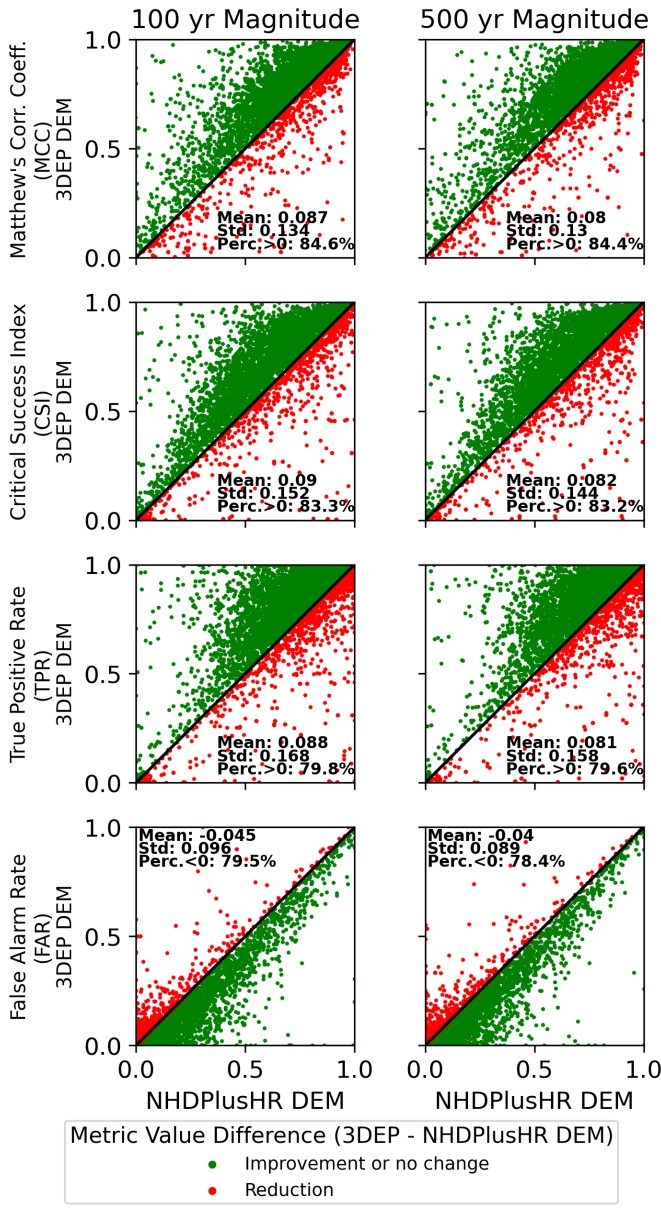

**Figure 6.** Figure shows catchment scale metric values. The eight sub-figures are organized by magnitude (100 and 500 yr) across the columns and for the four metrics across the rows. These values within each sub-figure are plotted on an axis representing HAND based FIMs generated from the NHDPlusHR DEMs (x-axis) and the same FIMs generated from 3DEP DEMs resampled to the 10 m spatial resolution (y-axis). The diagonal 45 degree line divides catchments that perform better with the legacy DEM (in red) from the catchments that perform better with the 3DEP DEM (in green). The majority of catchments perform better across all four metrics and both magnitudes with the higher quality 3DEP information. Additional descriptive statistics quantifying the distribution of metric differences (3DEP - NHDPlusHR) are also presented including the mean and standard deviation of the differences. We also included the percentage of samples whose difference is greater than or less than zero depending on the metric referenced.



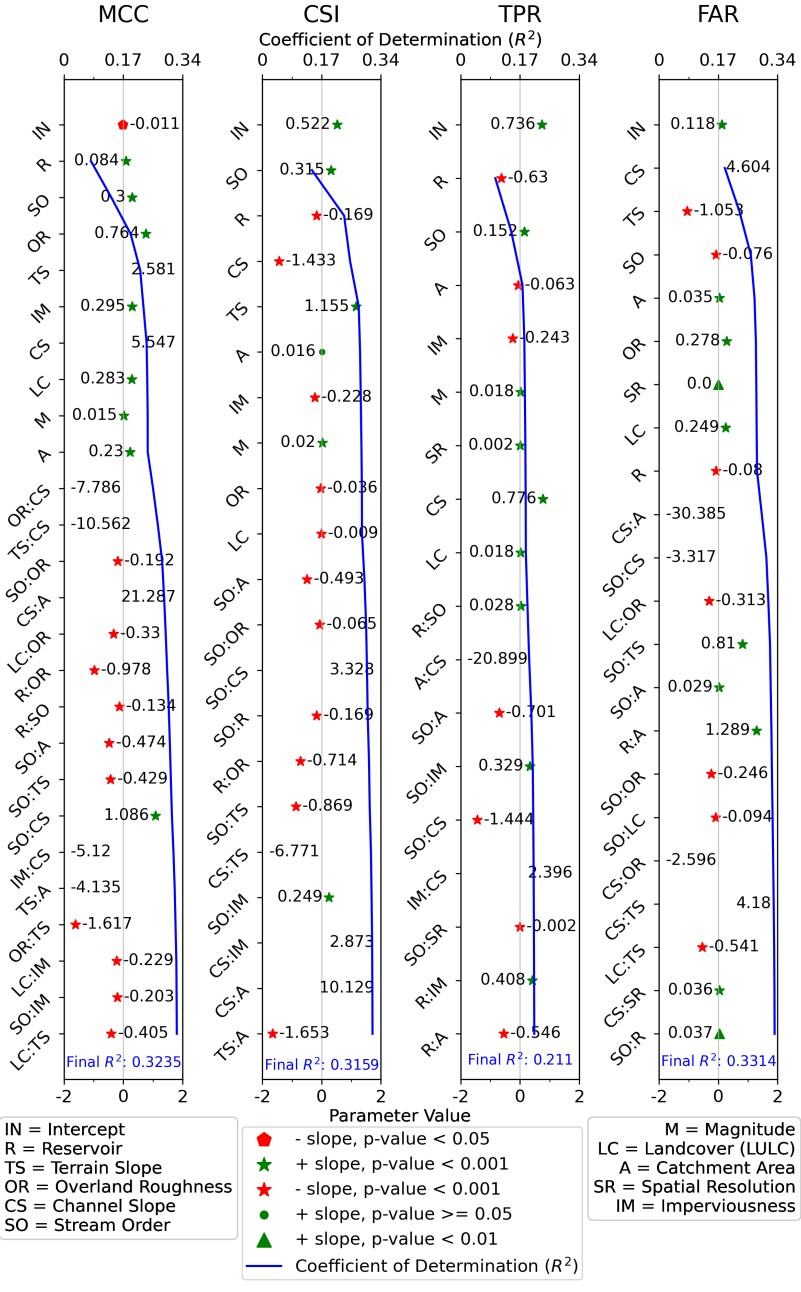

**Figure 7.** Figure illustrates the coefficients from multiple linear regression models fit on four response variables independently. The points on the graph represent the values of the coefficients while the shape represents the level of significance from $>= 0.05$ (circle), $< 0.05$ (pentagon), $< 0.01$ (triangle), and $< 0.001$ (star). The green and red colors represent the nature of the effects as either positive (direct) or negative (indirect), respectively. The models were built in a step wise fashion using forward model selection and AIC as a criteria for terminating the process. Eleven explanatory variables were considered for these models as well as their two-way interactions. An intercept was also included by default. As the models built, we recorded the $R^2$ of each successive model and tracked as the complexity of model increased. The final $R^2$ values for each final model are reported as well for each step of the forward selection.





Further examining Figure 7, we can infer interesting pieces of information as regression analysis is a tool to synthesize data into something interpretable. The coefficient of determination or $R^2$ values across the metrics vary from about 0.21 to 0.33. Translating this into other terms, one can say that about one fifth to one third of the catchment to catchment variance in the metrics can be explained by the eleven catchment scale, explanatory variables and their two-way interactions selected in this study. Additional, observation from the Figure illustrates the prevalence of certain explanatory variables near the beginning

of the selection process that seem to explain a fair amount of the variation as well as exhibiting strong effect sizes. Some of the variables of note include reservoir, stream order, terrain slope, channel slope, and LULC. These explanatory variables and there effect on catchment level performance on FIM will be examined later on in Section 3.4.

### 3.3    3DEP DEM Spatial Resolution

We investigated the effect of varying the spatial resolution of the 3DEP DEMs on the quality of FIMs produced from HAND.

The 3DEP DEMs were varied from 3, 5, 10, 15, and 20 m prior to HAND computation.

Figure 8 examines the relationship of DEM spatial resolution at five levels for each of the four metrics selected. The relationships are illustrated as distributions of catchment scale metric values for both event magnitudes (100 and 500 yr). We computed the distributions as kernel density estimations (KDEs) which is a non-parametric statistical technique that determines the probability distribution of a random variable. For each metric-magnitude distribution of catchment scale metrics, the 75th, 50th, and

25th percentiles are calculated and displayed from top to bottom as dotted, solid, and dotted lines, respectively. Additionally, we fit two linear regression lines, one for each magnitude, for all four metrics relating the linear effects of spatial resolution on metric values. The effect sizes, or the slopes of the regression lines, are displayed as well as their respective p-values. Low p-values denote effect sizes that are unlikely equal to zero.

Examination of Figure 8 shows statistically significant yet marginal in value effect sizes for the TPR and FAR metrics. For

example, the effect size of the TPR and 100 yr case is 0.0015 which represents a 0.0015 increase in the value of TPR for every unit m increase in the magnitude of the resolution. So for approximately 10 m, one would expect TPR to increase by 0.015. While coarser resolution DEMs appear to improve detection of inundation when compared to the BLE FIMs, it also appears have an undesirable effect on FAR as its expected values increase as DEMs are coarsened. These competing effects on TPR and FAR seem to have a canceling effect on the overall performance metrics of MCC and CSI. Both MCC and CSI both have

statistically insignificant trendlines which hints little to no overall improvement in catchment scale metrics of HAND based FIM by varying the spatial resolution of the input DEMs used to produce HAND.

Furthermore, we analyzed the mean and standard deviations of the inundated areas for the five spatial resolutions selected. Table 3 shows the HUC-8 level mean and standard deviation of inundated areas in km$^2$ by spatial resolution and across magnitudes. Very little variation in the inundated areas was seen across the resolutions which hints that while there was an

increase in TPR and FAR with coarser DEMs, there is also little change in the inundated areas. This suggests that most of the trade-offs in resolution were related to trading type I errors (FPs) in certain areas with type II errors (FNs) in other areas with little to no overall change in the inundated areas.





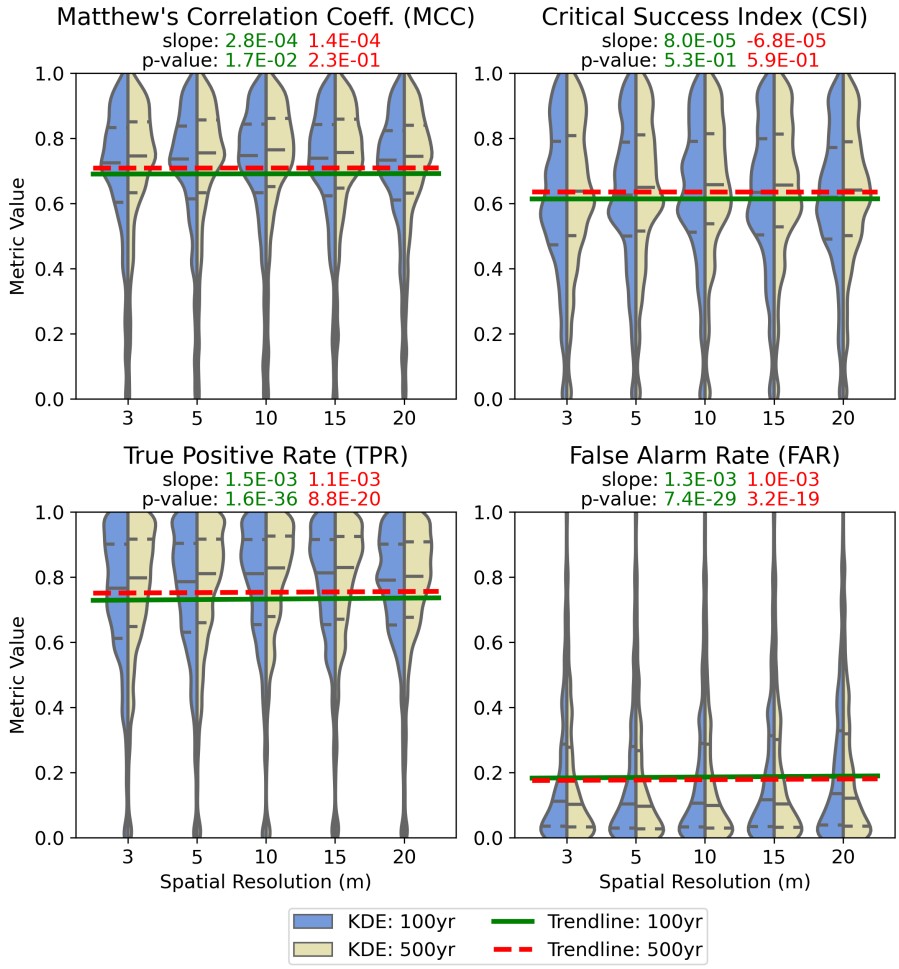

**Figure 8.** Illustrates the distribution of the four catchment scale metrics as violin plots across every spatial resolution selected including 3, 5, 10, 15, and 20 m. Each half of the violin represents a given magnitude of events (100 and 500 yr). Linear trendlines are fit for each metric-magnitude combination establishing linear relationships between spatial resolutions and metric values at the catchment scale.

**Table 3.** Mean and standard deviations of inundated areas across HUC-8's and magnitudes (100 and 500 yr in km$^2$ for each spatial resolution in meters.

| Spatial Resolution (m) | Mean Inundated Area (km$^2$) | Standard Deviation of Inundated Area (km$^2$) |
|:---:|:---:|:---:|
| 3 | 653.46 | 161.58 |
| 5 | 650.58 | 161.12 |
| 10 | 652.85 | 153.46 |
| 15 | 654.10 | 158.22 |
| 20 | 659.15 | 154.58 |





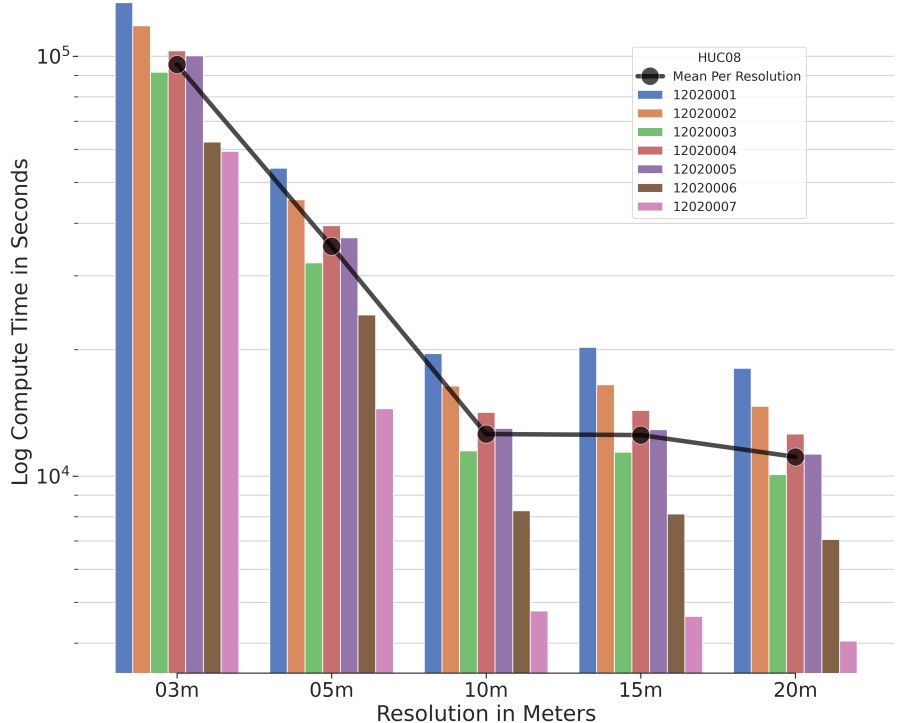

**Figure 9.** Total log CPU time in seconds across varying DEM spatial resolutions of 3, 5, 10, 15, and 20 m and listed by the seven HUC-8s in the study region. Resolution was found to have a significant effect on total CPU time for computing HAND as nearly an entire order of magnitude reduction in seconds was observed from changing the DEM resolution from 3 to 20 m.

A final observation related to the spatial resolution relates to its relative low importance or a lack of interaction variables in the models built for the regression analysis in Section 3.2 and Figure 7. This denotes that spatial resolution provided little to

no effect when considering impactful variables such as LULC, imperviousness, stream order, or reservoir.

DEM resolution was found to have a significant effect on the computational demands of producing HAND. We aggregated the times to compute HAND at the five various DEM spatial resolutions and found a significant effect on CPU time especially at finer resolutions. Figure 9 shows the change in log CPU times in seconds by HUC-8. An almost entire order of magnitude change in CPU time is seconds is observed when using DEMs of 3 versus 20 m resolutions. The number of pixels for a given

domain for squared pixels is known to have an inverse relationship with the square of spatial resolution ($numberOfPixels \propto resolution^{-2}$). So reducing spatial resolution from 10 m to 1 m represents a 100x increase in the number of pixels for the fixed domain. Despite having a minimal observed effect on skill, we found that higher resolutions tended to exhibit an excessive computational cost.



### 3.4 Explanatory Variable Focus

Since reservoirs and LULCs are valuable for forecasting operations, we elected to focus on those explanatory variables further within this analysis. Other variables, while important, were left out of scope for further analysis for this paper.

#### 3.4.1 Reservoirs

Given the relative importance of reservoirs in explaining catchment to catchment variance in the many of the metrics as shown in Figure 7 and Section 3.2, we isolate this factor out for further analysis here. Figure 10 shows the catchment level 455 distribution of the four metrics across spatial resolutions as violin plots built with KDEs. The halves of the violins are split across catchments that intersect NWM reservoirs and those that don't. The trendlines as well as their displayed slopes and p-values represent catchment scale metric variance as a function of spatial resolution for each reservoir group.

This Figure primarily shows a large statistical difference in catchment scale variation of three metrics, MCC, CSI, and TPR, across the catchments that intersect reservoirs and those that don't. Explaining this variation is simple as OWP FIM does not 460 currently account for reservoir related inundation while the BLE does. While the NWM reservoirs are currently masked out for evaluation purposes, the BLE reservoir inundation extents go beyond these masked regions thus contributing to FNs. Due to this fact, FAR illustrates very little performance difference across reservoir groups as FAR considers FPs and omits FNs (see Equation 3. Another important trend to denote from the previous Figure is the relative lack of interaction between spatial resolution and the reservoir factor shown by the similarity of the slopes of trendlines across reservoir groups. This can be 465 interpreted as spatial resolution having little effect across the reservoir groups which can also be seen in Figure 7 where the selection of a reservoir - spatial resolution predictor was omitted. Until OWP FIM accounts for reservoir flooding or some higher order masking technique is applied, the presences of reservoir related catchments will continue to contribute to a high variance in catchment scale metrics.

#### 3.4.2 Landuse/Landcover

We analyzed catchment scale metrics by taking the dominant landcover per catchment (mode). While the linear analysis in Section 3.2 grouped the NLCD categories into two groups depending on their degree of anthropogenic influence, we decided to ungroup the categories for Figure 11. In this Figure, we illustrate the distribution of the four catchment scale metrics shown as box plots which are grouped both by NLCD LULC and event magnitude. This chart doesn't appear to have a clear trend until further inspection leads one to see a pattern pertaining to the catchment scale agreement and the nature of the LULCs. To 475 reveal this trend, we decided to group LULC categories according to their relative level of anthropogenic influence.

In grouping the LULCs by two categories of "more" and "less" anthropogenic influence we are able to see a clearer trend as to how LULC affects catchment scale agreement. The LULCs grouped into the "more" category include the developed categories (open space, low intensity, medium intensity, and high intensity) and the cultivated crops category which depending on the cropping system can have significant hydrological implications. The remaining LULCs within the study area were 480 placed in the "less" category. Figure 12 shows the distribution of catchment scale metrics sorted by grouped LULC and event

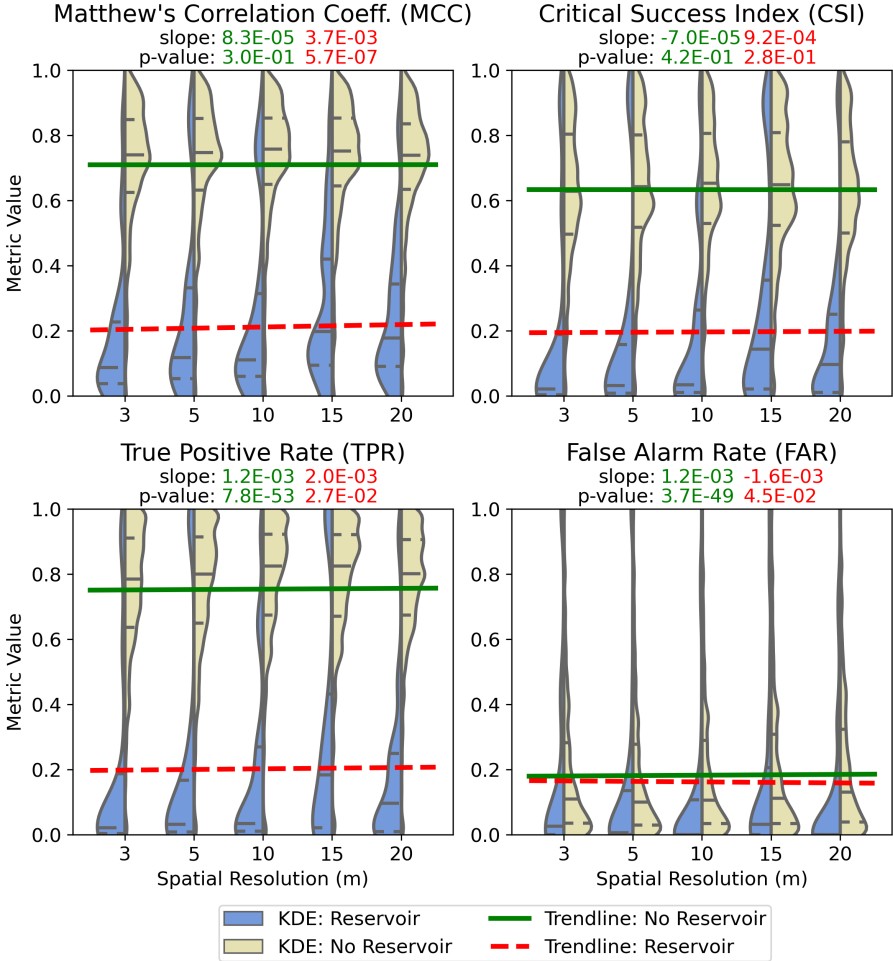

**Figure 10.** Catchment scale variation illustrated as distributions modeled as KDEs. The distributions are grouped by metric and by spatial resolutions. The halves of the violins are divided by the presesence of a NWM catchment that intersects a NWM reservoir or not. Significant differences are observed between catchments identified with reservoirs and those that are not for all resolutions and metrics employed. Reservoirs are not currently modeled within OWP FIM while the BLE does account for reservoir related inundation extents. While the NWM reservoirs are masked out for evaluation purposes, some of the BLE inundation extents reach beyond these boundaries leading to significant amount of FNs. The trendlines as well as their corresponding slopes and p-values were constructed by regressing the two reservoir groups independently on spatial resolution. Little to now interaction between reservoir groups and spatial resolutions was observed.



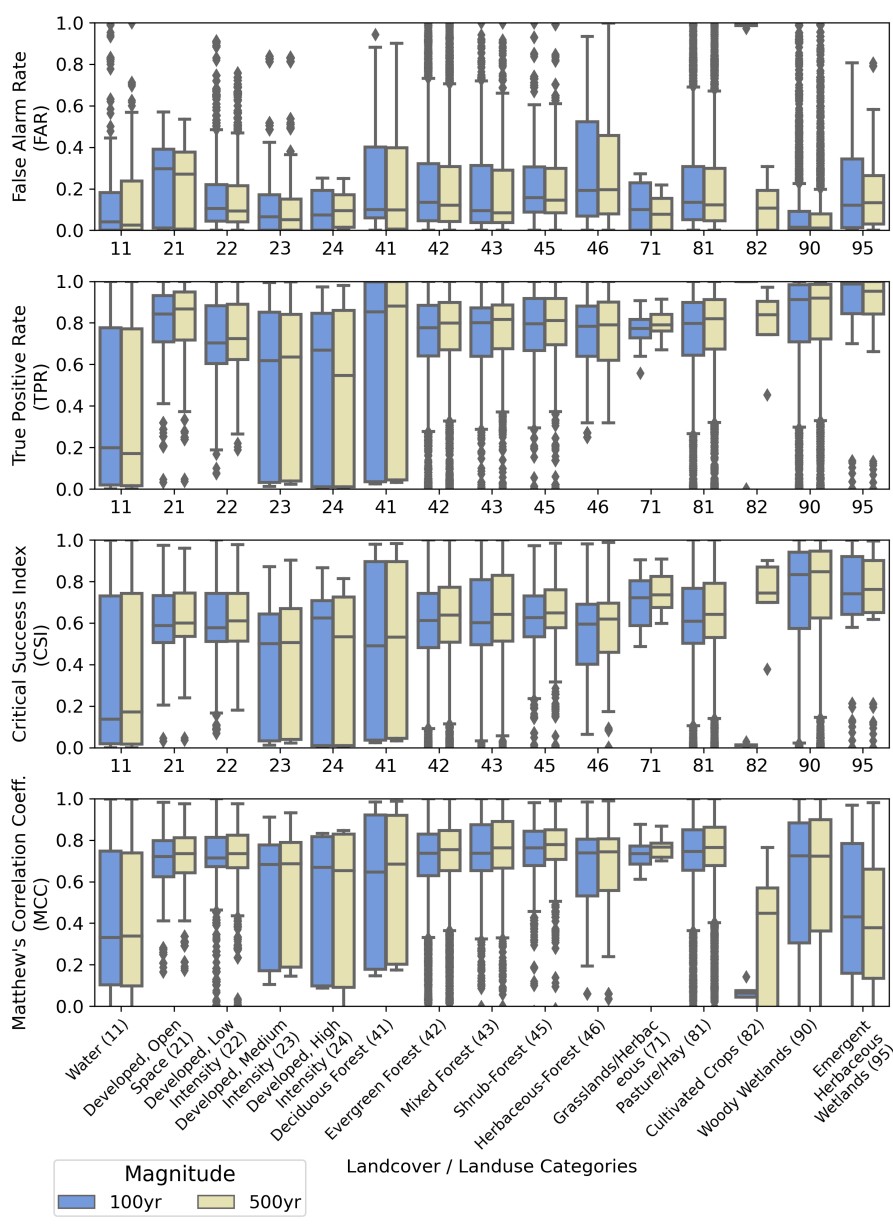

**Figure 11.** The distribution of four catchment scale agreement metrics are shown as box plots and grouped by the dominant NLCD LULC per catchment as well as the event magnitude.



magnitude. We fit a multiple linear regression model for each metric using the grouped LULC and magnitude as factors as well as their interaction. The resulting formulas for this linear modeling are shown above each Figure with the parameter values and their relative level of significance. Since only p-values greater than 0.05 and less than 0.001 were encountered, we denoted those with no asterisk and 3 astericks, respectively. Additionally, we plot the trendlines resulting from another regression that associates the metric values to the LULC grouping and does this for each event magnitude independently. Illustrating this regression demonstrates these relationships in a qualitative manner highlighting the lack of interaction of event magnitude and grouped LULC.

Judging from the Figures 11 and 12, there is a clear indication that LULC has a significant influence over catchment scale agreement. Grouped LULCs in Figure 12, show the importance of anthropogenic influence on explaining catchment scale variation in metric values with a negative relationship observed for having "more" relative anthropogenic influence. We found that LULC affected all the metrics except for FAR where over-prediction was found not be as affected by the anthropogenic influence. Under-prediction does appear prevalent in regions of anthropogenic influence which could be explained by a variety of factors including DEM inconsistencies or adverse affects on hydro-conditioning in areas with rapidly varying or uncertain elevations. It does appear that anthropogenic influence also contributes to more variation within the "more" case than the "less" case which could be a result of noise that is inherited from elevation inputs. While the magnitude per se is a significant factor in explaining catchment scale agreement, it does not interact with grouped LULC meaning anthropogenic influence seems to carry a similar effect across event magnitudes. Another interesting observation related to LULC is that the grouped LULCs don't seem to interact with spatial resolution. So for this study area, higher resolutions did not provide an improvement in metrics for regions with more anthropogenic influence. We leave further analysis of the effect of LULC and the anthropogenic influence on catchment scale agreement to future work.

## 4 Discussion

Our results and analysis demonstrated several key methods that can improve the agreement of continental scale FIM using HAND when compared to engineering scale FIM models. The inclusion of higher quality terrain information from 3DEP was able to significantly improve the quality of continental scale HAND based FIMs. This finding is consistent with previous studies that have come to similar conclusions (Li et al., 2022b; Zheng et al., 2018a; Garousi-Nejad et al., 2019; Speckhann et al., 2018), while also meeting the goal of the 3DEP objectives set out in justifying the collection effort (Dewberry, 2011; Snyder et al., 2013; Sugarbaker et al., 2014). As the program approaches continental scale availability, 3DEP data can be justified for use directly for HAND computation for the entire US leading to enhanced FIM and forecast quality.

However, varying the spatial resolution of the 3DEP DEMs from their native 1 m was found to have little effect on the quality of HAND based FIMs. Previous studies examining this (Li et al., 2022b; Zheng et al., 2018a; Garousi-Nejad et al., 2019; Speckhann et al., 2018) have varied in their experimental design and modeling assumptions. The modeling assumptions in those studies were different than those used in our HAND methods as we employ different datasets, hydro-conditioning procedures, and compute HAND at finer scales (Aristizabal et al., 2022). The experimental design of some previous studies







**Figure 12.** Distribution of catchment scale agreement across four metrics illustrated as box plots and grouped by the level of anthropogenic influence in the NLCD LULCs and the two event magnitudes. The results of a linear model that regress the catchment scale metrics on grouped LULC, magnitude, and their interaction are shown. The coefficients for the model are labeled with their p-values by no asterisk and three astericks for the p-values greater than 0.05 and less than 0.001, respectively. Additionally the trendlines resulting in a regression with catchment scale metrics on grouped LULC are shown per event magnitude.





(Zheng et al., 2018a; Garousi-Nejad et al., 2019) looked at high resolution terrain data but did not explicitly isolate that factor

out for analysis purposes. Additionally, previous studies failed to denote a consistent relationship with spatial resolution and

FIM performance (Li et al., 2022b; Speckhann et al., 2018). While the mechanisms of this relationship have not been thoroughly

explored with HAND, others have found that spatial resolution may have a spurious relationship with FIM performance due

to inherent uncertainties related to this problem (Savage et al., 2016). Future research can expand the analysis of spatial

resolution's effect on FIM quality to more study sites across broader domains of interest. Future research can also explore the

effect spatial resolution may have on the quality of FIM depths as they likely behave somewhat independently to extents.

Further analysis sought to explain some of the catchment level variation in the four agreement metrics with the aim of

indicating where future progress can be made in extending FIM quality for continental scale applications. As a result of the

regression analysis, a fifth to a third of the catchment scale variation of the agreement metric values was explained by building

linear models with eleven explanatory variables and their two-way interactions. These models, while used for analysis purposes

in this study, can have predictive performance which could have calibration applications. Previous works have used a variety

of methods to help calibrate Manning's $n$ or bathymetry (Zheng et al., 2018b; Johnson et al., 2019; Jian et al., 2017; Neal

et al., 2021; Liu et al., 2019). Due to the complex and interconnected nature of the source hydrography, hydro-conditioning

operations, reach-averaged channel geometry, and Manning's n, any sort of calibration for SRCs would involve using the same

sets of methods and datasets used to produce our version of HAND based FIM.

Some of the explanatory variables were explored in further detail to provide insights on future possible skill improvements

for OWP HAND based FIM. Reservoirs were found as one of the leading independent variables in explaining catchment scale

variation in three of the four metrics mostly driven by under-prediction or FNs. These errors are caused by not accounting

for reservoir inundation within OWP HAND FIM. Several methods exist for accounting for reservoir inundation which could

leverage volume computations with the NWM (Gochis et al., 2021; Chen et al., 2018; Shin et al., 2019). It's important to

note here that while reservoirs explained a significant amount of variation in the metrics, it does not mean that accounting for

reservoir inundation properly would lead to a significant increase in agreement. Agreement will only change in response to

the quality of the new method employed as well as the prevalence of reservoir inundation in a given region. Another variable

of interest further analyzed included the effect of LULC on agreement metrics. We found that HAND FIMs did not perform

as well in catchments that are labeled as developed or cultivated crops. Regions of high anthropogenic influence negatively

influence the performance of inundation models by adding extra complexity within the terrain information as well as with

the physics employed. Furthering performance in these regions could benefit from use of hyper-resolution models that better

account for urban water features (Grimley et al., 2017; Smith et al., 2020; Deo et al., 2018; Gurung et al., 2018; Smith et al.,

2021; Leandro et al., 2016; Chegini et al., 2021). Further exploring these and other independent variables could help inform

future development directions to help improve the quality of continental scale FIM techniques.



## 5 Conclusions

Floods are a significant source of natural disasters in the United States (US) leading to loss of property and lives. The National Oceanic and Atmospheric Administration (NOAA)'s Office of Water Prediction (OWP) has implemented a National Water Model (NWM) to help forecast streamflows at nearly three million locations across the continuous US, Hawaii, Puerto Rico, and portions of Alaska at hourly time steps and multiple forecast horizons. The OWP has developed its own version of Height Above Nearest Drainage (HAND) that accounts for multiple fluvial sources of inundation instead of just that from the local, nearest flowpath. The United States Geological Survey (US Geological Survey)'s 3-Dimensional Elevation Program (3DEP) is rapidly approaching continental scale availability so we evaluated its use at the 10 meter (m) spatial resolution within the derivation of HAND and found significant increases in the quality of flood inundation map (FIM) performance. Additionally, we varied the resolution to include 3, 5, 15, and 20 m but did not find any significant trends on an overall basis leading one to justify its use within this study regions. As one would expect, the computational time increased to compute HAND with the number of digital elevation model (DEM) cells considered, which goes up with the inverse of the DEM resolution squared. We determined that more studies are required in other regions to help explore the potential benefits with the use of higher resolution DEMs along with HAND. A multiple linear regression model fitting eleven factors and covariates to the four agreement metrics all at the catchment scale revealed that about one fifth to one third of the variation can be explained by these explanatory variables.

*Code and data availability.* HAND data produced for this study can be found on our Earth Science Information Partners (ESIP) backed Amazon Web Services S3 bucket (Aristizabal et al., 2023a). Software used in this study is available on GitHub (Aristizabal et al., 2023b). A permanent version of this code and data is available as well. (Aristizabal, 2023).

*Author contributions.* Conceptualization, FA and FS; methodology, FA, FS, and JJ; software, FA and TC; validation, FA; formal analysis, FA and FS; investigation, FA and FS; resources, FS; data curation, FA and TC; writing—original draft preparation, FA and TC; writing—review and editing, FA, FS, TC, and JJ; visualization, FA; supervision, FS and JJ; project administration, FS and JJ; funding acquisition, FS All authors have read and agreed to the published version of the manuscript.

*Competing interests.* The authors declare that they have no conflict of interest.

*Acknowledgements.* We would like to thank Jason Stoker and his team at the 3DEP for producing analysis friendly services and for answering our questions related to these services. Additionally, we would like to thank Fred Ogden, chief scientist of the Office of Water Prediction



(OWP), for his help reviewing and providing scientific direction. We would like to acknowledge the use of large language models (LLMs) in the editing process of this manuscript to enhance the clarity, coherence, and grammatical accuracy of our writing.



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
