# Peer review of "Effects of High-Quality Elevation Data and Explanatory Variables on the Accuracy of Flood Inundation Mapping via Height Above Nearest Drainage"

_EGUsphere, 2023_

## Author Comment (AC1)

**Reviewer 1**

**Introductory Response**

Dear Reviewer,

We sincerely thank you for your constructive feedback and the positive remarks regarding the writing, methodology, and interpretation of our manuscript. We are encouraged by your recognition of the importance of our work in exploring the uncertainties and improving the accuracy within the National Oceanic and Atmospheric Administration (NOAA) Office of Water Prediction (OWP) flood inundation map (FIM) framework.

We understand the significance of the issues you have highlighted and are fully committed to addressing these to further enhance the quality and impact of our manuscript. Our primary objective remains to explore and elucidate the effects of digital elevation model (DEM) source and resolution on FIM extents, thereby contributing valuable insights toward improving the accuracy and reliability of the OWP Height Above Nearest Drainage (HAND) FIM framework for operational forecasting.

We will address each of the issues you have highlighted in the subsequent sections of our response letter, providing clarifications and making necessary revisions as suggested. We appreciate the opportunity to enhance our manuscript through this revision and are committed to making all requisite amendments to meet the high standards of Hydrology and Earth System Sciences (HESS).

Thank you once again for your insightful feedback.

Warm regards,

Corresponding Author

**Major Issues**

**Reviewer Point P 1.1** — Consider extending the analysis for 30 and 90 meter (m) resolutions - these are the most common DEM resolution for many national and global DEMs.

**Reply**: We appreciate your suggestion to extend the analysis to include coarser resolutions, as these are commonly encountered in many national and global scale DEMs. Additionally, we believe that this extension can provide valuable insights into answering at what point the coarsening of resolutions begins to affect the quality of HAND based FIMs significantly.

In acknowledgment of this valuable input, we have conducted additional analyses and expanded our discussion in Section 3.3 to encompass these additional spatial resolutions. The outcomes of this extended analysis have been encapsulated in a newly introduced Table 5, which illustrates how the quality of FIM begins to degrade at very coarse resolutions.

These amendments aim to provide a broader understanding of how different DEM resolutions impact the FIM quality, aligning with other resolutions utilized in national and global scale DEMs. We trust that this inclusion enriches our manuscript and addresses the concern adeptly.

**Reviewer Point P 1.2** — Figure 4. A very busy map that makes it hard to read - consider aggregating the key classes into 4-5 major land use/land cover (LULC) classes.

**Reply**: Thank you for your constructive suggestion to simplify Figure 4 by aggregating key classes into fewer major LULC categories. In response to your feedback, we have revised the figure by reducing it to the high-level National Land Cover Database (NLCD) LULCs classes, which has indeed made the figure less crowded and more comprehensible.

The revised figure now predominantly presents four main land use/land cover classes: forest, wetland, open-water, and pasture/crops, providing a clearer visual representation. Some high-level indication of anthropogenic development is also provided by this new symbology. We believe that this revision provides a more straightforward visualization while retaining the essential information for understanding the land cover distribution across the site.

We trust that this amendment addresses your concern effectively and enhances the clarity and readability of Figure 4 in our manuscript.

**Reviewer Point P 1.3** — Section 3.1 and Figure 6: I found it hard to understand what is the actual/overall improvement in HAND-FIM predictions using 3-Dimensional Elevation Program (3DEP). The authors should provide summary statistics in a table and/or additional plots (e.g. PDF, Box Violine).

**Reply**: We appreciate your valuable feedback regarding the presentation of the improvement in HAND-FIM predictions using 3DEP in Section 3.1 and Figure 6. We understand the significance of clearly conveying the actual improvement to elucidate the broader impact of our study.

While the original Figure 6 aimed to explicitly depict the difference in metric values, we acknowledge your suggestion for a more comprehensive presentation of the summary statistics. To address this, we have included summary metrics within a new table—Table 5, which presents the mean catchment scale metric values across various DEM sources and spatial resolutions, while aggregating across flood magnitudes. Also, we removed the differences as to focus on the distribution of the metrics values in Figure 6 and not repeat information.

We trust these amendments will provide a clearer and more thorough understanding of the improvements achieved through the use of 3DEP, thereby addressing the concerns you raised.

**Reviewer Point P 1.4** — Section 3.2 and Figure 7: this was the most unclear section of the manuscript. The results did not make much sense to me and the figure was hard to read/understand. The authors should re-think how to present and analyze the results. They may want to consider removing this section altogether

**Reply**: Thanks for pointing out some of the opportunities to improve the regression analysis results. We understand this analysis technique can appear unclear. To better align with established practices for conveying multivariate regression analysis, we included a new table-Table 3 in order to convey the slopes and their levels of significance better showing the catchment scale metrics and how they relate to various explanatory variables. We hope that this table better aligns with traditional practices in reporting and offers our audience an easier time in understanding the meaning of this analysis.

**Reviewer Point P 1.5** — Section 3.3 and Figure 8: I don't recall that the authors explained how these distributions were calculated. Similar to section 3.1, summary (overall) statistics of the FIM accuracy metrics should be reported.

**Reply**: We appreciate your insightful observation regarding the clarification of the distributions calculation in Figure 8, and the request for overall summary statistics of the FIM accuracy metrics.

In the initial manuscript, the calculation of distributions using Gaussian kernel density estimation (KDE) was briefly mentioned on lines 417-419 of Section 3.3. To address your concern, we have revised these lines to provide a more detailed explanation of the Gaussian KDE methodology employed, along with appropriate references to ensure a comprehensive understanding for the readers.

Furthermore, in alignment with your suggestion and similar to the amendment made in your previous Point P 1.3, we have included overall summary statistics of the FIM accuracy metrics in the newly introduced Table 5. This table aims to provide a clear and concise summary of the metrics, facilitating a better understanding of the accuracy improvements.

We trust that these revisions aptly address your comments and enhance the clarity and completeness of our manuscript in the discussed sections.

**Reviewer Point P 1.6** — Figure 9 and relevant text: report the computer hardware that was used.

**Reply**: We thank the reviewer for bringing to our attention the importance of reporting the computer hardware used in our analyses, as this information is crucial for the reproducibility and comprehension of the computational benchmarks presented.

In response to your valuable comment, we have now included a detailed description of the computer hardware and the operating system utilized for computing these benchmarks in both the caption of Figure 9 and the corresponding text section. This addition will provide readers with a clearer understanding of the computational environment in which our analyses were conducted.

We trust that this amendment adequately addresses your concern and enhances the transparency and reproducibility of our work.

**Minor Issues**

**Reviewer Point P 1.7** — Line 35: '...scales [often] requires'

**Reply**: Addressed.

**Reviewer Point P 1.8** — Line 109: 'omb' ?

**Reply**: We appreciate your keen observation regarding the unclear citation on line 109. The issue arose from how BibTeX renders citations with institutional authors. We have now rectified this problem by ensuring that the full institution name, Office of Management and Budget, is accurately cited in the text. This amendment will eliminate any confusion and enhance the clarity of the citation.

**Reviewer Point P 1.9** — Line 168: 'due to the (...'

**Reply**: Thank you for bringing to our attention the incomplete statement on line 168. We have reviewed and revised the sentence to convey and complete the intended meaning better. We trust this revision addresses your concern and enhances the clarity of the manuscript.

**Reviewer Point P 1.10** — Line 183: '?'

**Reply**: This question mark was introduced by BibTex due to a repeated citation reference. This repeated entry has been removed and the question mark has disappeared. Thank you for you careful attention to detail.

**Reviewer Point P 1.11** — Line 256: '(of DEMs ,...'; '(, WBM)...'

**Reply**: Another issue related to BibTex rending of institutions. Thank you for your close oversight. This issues has been addressed.

**Reviewer Point P 1.12** — Line 257: '(, MRLC)'

**Reply**: Related to previous point P 1.11. This is now addressed.

**Reviewer Point P 1.13** — Line 373: 'we used investigate...'

**Reply**: The sentence has been rephrased for clarity and grammatical correctness.

**Reviewer Point P 1.14** — Line 400: 'two-way interactions' - do you mean cross-correlation?

**Reply**: Thank you for bringing up this point. In line 400, when we mention "two-way interactions," we are referring to interaction terms within the regression model, where the combined effect of two explanatory variables on the response variable is examined. This is common terminology in regression analysis to denote interactions between different predictor variables. The term "cross-correlation" generally pertains to the relationship between two separate series of data, which is not the focus in our regression analysis.

To ensure clarity and avoid any confusion, we will add a brief explanation of the "two-way interactions" in the text surrounding line 400.

We appreciate your vigilance in ensuring clarity and precision in our manuscript.

**References**

---

## Author Comment (AC2)

**Reviewer 2**

**Introductory Response**

Dear Reviewer,

Thank you for your valuable and thorough feedback. We are grateful for your positive remarks concerning the organization, relevance, and topicality of our paper, as well as your appreciation for the importance of the outcomes depicted in Figure 6.

We acknowledge your concern regarding the generality of our results, given that they are based on a single study area with 'low terrain slope and minimal anthropogenic influence.' We understand that the scope of inference could be broadened with a more varied selection of study areas in future works as National Oceanic and Atmospheric Administration (NOAA)'s Office of Water Prediction (OWP) continues to expand its coverage regions. The unique characteristics of the study area were chosen to minimize extraneous variables and to allow a focused examination of the digital elevation model (DEM) quality's impact. Nonetheless, we see the benefit of including more diverse terrain and environmental conditions in future analysis to make the findings more generalizable.

We also recognize your comments on the less intriguing results regarding the role of different spatial resolution on flood inundation map (FIM) detection and the explanatory factors and co-variates' capacity in explaining the variance of the obtained results. We value this observation and plan to delve deeper into these aspects, aiming to provide a more comprehensive understanding and clearer exposition of these relationships in future works.

Further, we are committed to revisiting the analyses and expanding on the discussions to address the concerns you've raised, with a particular focus on enriching the examination of spatial resolution effects and the explanatory power of different factors and covariates on the metrics' performances.

We sincerely appreciate your insightful feedback, which provides us with a clear pathway to enhance the quality and breadth of our manuscript.

Thank you once again for your time and constructive critique.

Warm regards,

Corresponding Author

**Major Comment**

**Reviewer Point P 2.1** — The most important perplexity in my reading of the paper regards the choice of the basic source of information for both evaluation and validation of the method, which is a 1-Dimensional (1D) Hydrologic Engineering Center River Analysis Center (HEC-RAS) flood inundation extents, involving hydrologic and 1D hydraulic of Saint Venant equations.

**Reply**: Thank you for your insightful comment regarding the choice of benchmark data for evaluating and validating our method. We recognize that the selection and application of validation data are pivotal to the conclusions drawn in our study. Evaluating FIM quality poses a significant challenge due to data sparsity, quality, and uncertainty inherent in available benchmark datasets. Common techniques for evaluating large-scale, low complexity FIM models—including high-water marks, remote sensing, in-situ

gauges, crowd-sourced observations, damage assessments, and higher-order models—all have limitations and uncertainties that could potentially impact the validity of conclusions drawn.

Your comment has sparked a detailed examination, leading us to segment the discussion into three main components: Base Level Engineering (BLE) cross-section usage, BLE benchmark DEM quality and resolution, and the suggestion of employing 2-Dimensional (2D) hydraulic modeling alongside high-resolution DEMs in future work.

In the following points, we delve into these components, hoping to clarify our rationale and express our willingness to explore the suggestions you have proposed for future studies.

**Reviewer Point P 2.2** — From the reading it seems that the BLE cross sections provided by Interagency Flood Risk Management (InFRM) were used both for comparison of FIMs (validation) and for the evaluation of the Height Above Nearest Drainage (HAND) metric. If this is true, I believe the authors should better specify the reason of such a choice.

**Reply**: We appreciate your observation regarding the utilization of BLE cross-sections in our analysis. These cross-sections are indeed central to our evaluation of HAND based FIMs across different DEM sources and resolutions.

The BLE cross-sections are valued for their inclusion of geo-referenced streamflow values, serving as crucial forcing information for HAND based synthetic rating curves (SRCs) and FIMs. By spatially intersecting these cross sections with the forecasting stream network of the National Water Model (NWM) we could generate reach level streamflows for mapping, aiding in a more precise evaluation.

These values are the same streamflow values used in generating the rating curves and extents within the BLE allowing for parity and fair comparison of stage-discharge relationships and mapping techniques. This approach allowed us to bypass the use of streamflow runoff models such as the NWM, which could potentially introduce additional hydro-climatic errors and uncertainties. It offered a more controlled evaluation scenario compared to using in-situ stream gages, which are often sparse and may not provide a comprehensive basis for validation.

While we introduced this methodology and its rationale in the manuscript, a more elaborate justification is provided in previous works (Aristizabal et al., 2023). To enhance clarity, we reiterated some key points from this reference in Section 2.6 of our manuscript.

We trust this response, along with the explanations provided in the manuscript and referenced work, clarifies our methodological choices. We are open to further elaborating on this approach in the manuscript to ensure a clear understanding of our validation strategy.

**Reviewer Point P 2.3** — I may imagine that such a choice can make sense in the aim of a larger comparison at the continental scale, on the other hand I think that it is hard to look for improvements based on higher quality and higher resolution DEM, whenever the benchmark has not the same quality.

**Reply**: Given the central thesis of our manuscript revolves around investigating DEM sources and resolutions alongside their effects on FIM quality, it is crucial to address the realities inherent within the benchmark dataset to uphold the integrity of our conclusions.

Upon receiving this comment, we revisited the BLE documentation to ascertain the suitability of these datasets in testing our thesis. The documentation elucidates that high-quality Light Detection and Ranging (LiDAR) data covers the entire Hydrologic Unit Code (HUC)-4 region, 1202, used as our study area and that this DEM at a 10-foot resolution was employed for hydraulic analysis and floodplain

mapping within the BLE (*Base Level Engineering Analysis: Region 6 Neches River Watershed – Lower Angelina (HUC8 - 12020005)*, 2019; *Base Level Engineering Analysis: Region 6 Neches River Watershed – Lower Neches (HUC8 - 12020003)*, 2019; *Base Level Engineering Analysis: Region 6 Neches River Watershed – Middle Neches (HUC8 - 12020002)*, 2019; *Base Level Engineering Analysis: Region 6 Neches River Watershed – Pine Island Bayou (HUC8 - 12020007)*, 2019; *Base Level Engineering Analysis: Region 6 Neches River Watershed – Upper Angelina (HUC8 - 12020004)*, 2019; *Base Level Engineering Analysis: Region 6 Neches River Watershed – Upper Neches (HUC8 - 12020001)*, 2019; *Base Level Engineering Analysis: Region 6 Neches River Watershed – Village (HUC8 - 12020006)*, 2019).

Affirmation of the utilization of high-quality, LiDAR-derived DEMs in the benchmark data lends further credibility to the positive outcomes depicted in Figure 6. Moreover, it furnishes us with additional validation that this dataset facilitates the examination of various spatial resolutions down to 1 meter (m) if feasible. It also reinforces our assertion that higher resolutions do not invariably enhance the quality of HAND based FIMs across all regions, necessitating further research to ascertain the utility of higher resolutions in varying regions, especially given their significant computational expense.

Following your comment, we have enriched the introductory paragraph of Section 2.6 to ensure the reader is well-informed about the high-resolution, high-quality LiDAR-derived DEMs employed within the BLE benchmarks. We trust that this response, coupled with our revisions to the manuscript, adequately addresses your comment and further augments the quality of our manuscript.

**Reviewer Point P 2.4** — I would suggest, maybe for future works, a comparison with flood inundation maps obtained with a shallow-water complete 2D hydraulic modeling supported by high resolution DEMs. Another feasible analysis could be carried out over real inundation maps.

**Reply**: Your suggestion raises a crucial aspect of advancing our large-scale, low-complexity FIM model by juxtaposing it with higher-order physics in subsequent investigations. To resonate with this point, we have enriched our Discussion section, articulating the potential of incorporating 2D hydraulic modeling in future research as benchmark datasets. We trust that this revision aptly addresses your insightful suggestion and amplifies the depth of discussion within our manuscript.

**Minor Issues**

**Reviewer Point P 2.5** — The sentence in lines 320-321 seems not consistent with equation (1).

**Reply**: Thank you for your careful observation. We have adjusted the mentioned sentence for overall clarity and for consistency with equation (1). We hope this change addresses your concern.

**Reviewer Point P 2.6** — Lines 343 (and around it) it is not clear the difference between covariates and factors. The different role they play in the regression analysis and also how their combination are made.

**Reply**: Thank you for bringing this to our attention. We acknowledge that the terminology and the methodological explanation around covariates and factors need to be clarified. In the revised manuscript, we have elucidated these terms and their roles in the regression analysis as follows:

Factors are categorical variables in our analysis that have a finite number of distinct categories. They are utilized to group the data into different levels, each representing a category. For example, grouped

land use/land cover (LULC) could be a factor with levels of anthropogenic influence such as 'more' or 'less'.

Covariates, on the other hand, are continuous variables that are assumed to have a linear relationship with the dependent variable. For example, channel slope could serve as covariates in our analysis.

In our regression analysis, we included both covariates and factors to capture the variance in the dependent variable more comprehensively. The combination of covariates and factors was carried out by including interaction terms in the regression model. Interaction terms are created by multiplying a covariate and a factor or two factors, which allows us to investigate whether the effect of one variable depends on the level of the other variable.

We hope that this explanation clarifies the different roles of covariates and factors, and the methodology of combining them in our regression analysis. We have augmented Section 2.6.1 which introduces these terms to ensure they are properly conveyed to the interested reader.

**Reviewer Point P 2.7** — It is not clear if and/or how the spatial resolution of DEM affects results in figures 11 and 12.

**Reply**: Your observation regarding the potential interaction between spatial resolution and LULC is insightful. We, too, recognized that there could be a significant interaction between these variables, which prompted us to conduct a regression analysis to explore this possibility further. The outcomes of this analysis, primarily depicted in Figure 7, reveal that the interaction between spatial resolution and stream order is only minimal in only the case of true positive rate (TPR), indicating that in our study, spatial resolution does not significantly interact with other variables. This observation was initially mentioned in lines 497-500, underlining the lack of interaction between spatial resolution and LULC in affecting FIM quality. To clarify this aspect further, we have enhanced the caption of Figure 12 to explicitly state that our analysis found no significant interaction between LULC and spatial resolution.

**References**

Aristizabal, F., Salas, F., Petrochenkov, G., Grout, T., Avant, B., Bates, B., ... Judge, J. (2023). Extending height above nearest drainage to model multiple fluvial sources in flood inundation mapping applications for the us national water model. *Water Resources Research*, e2022WR032039.

*Base level engineering analysis: Region 6 neches river watershed – lower angelina (huc8 - 12020005)* (MIP Deliverable No. 16-09-0654S). (2019, August). Strategic Alliance for Risk Reduction II (STARRII). (FEMA IDIQ Contract: HSFE60-15-D-0005)

*Base level engineering analysis: Region 6 neches river watershed – lower neches (huc8 - 12020003)* (MIP Deliverable No. 16-09-0654S). (2019, August). Strategic Alliance for Risk Reduction II (STARRII). (FEMA IDIQ Contract: HSFE60-15-D-0005)

*Base level engineering analysis: Region 6 neches river watershed – middle neches (huc8 - 12020002)* (MIP Deliverable No. 16-09-0654S). (2019, August). Strategic Alliance for Risk Reduction II (STARRII). (FEMA IDIQ Contract: HSFE60-15-D-0005)

*Base level engineering analysis: Region 6 neches river watershed – pine island bayou (huc8 - 12020007)* (MIP Deliverable No. 16-09-0654S). (2019, August). Strategic Alliance for Risk Reduction II (STARRII). (FEMA IDIQ Contract: HSFE60-15-D-0005)

*Base level engineering analysis: Region 6 neches river watershed – upper angelina (huc8 - 12020004)* (MIP Deliverable No. 16-09-0654S). (2019, August). Strategic Alliance for Risk Reduction II (STARRII). (FEMA IDIQ Contract: HSFE60-15-D-0005)

*Base level engineering analysis: Region 6 neches river watershed – upper neches (huc8 - 12020001)* (MIP Deliverable No. 16-09-0654S). (2019, August). Strategic Alliance for Risk Reduction II (STARRII). (FEMA IDIQ Contract: HSFE60-15-D-0005)

*Base level engineering analysis: Region 6 neches river watershed – village (huc8 - 12020006)* (MIP Deliverable No. 16-09-0654S). (2019, August). Strategic Alliance for Risk Reduction II (STARRII). (FEMA IDIQ Contract: HSFE60-15-D-0005)

---

## Author Response (AR2)

**Reviewer 1**

**Introductory Response**

Dear Reviewer,

We sincerely appreciate your dedication and time spent reviewing our manuscript across both this and previous rounds of feedback. Your comments have been instrumental in enhancing the quality and clarity of our work. It is gratifying to know that our recent revisions have successfully addressed your earlier concerns.

In this round of revisions, we have meticulously ensured that each of your comments has been thoroughly considered and addressed, aligning our manuscript with the high standards expected by HESS.

With gratitude,

Corresponding Author

**Minor Issues**

**Reviewer Point P 1.1** — Updating the abstract and conclusion with the additional analyses and results.

**Reply**: The abstract and conclusion sections have been comprehensively revised to incorporate the recent analyses and findings, thereby ensuring that they accurately reflect the expanded scope and insights of our research.

**Reviewer Point P 1.2** — New digital elevation model (DEM) resolution runs not added to Figure 9

**Reply**: We have duly updated Figure 9 to include the new DEM resolution runs. This modification provides a more detailed and complete representation of the results, aligning with the latest advancements in our analysis.

**Reviewer Point P 1.3** — Formatting issues - 'spillover' references and Table 3.

**Reply**: We acknowledge the formatting concerns raised. However, the issues noted, specifically the 'spillover' references and the layout of Table 3, are artifacts of the limitations within latexdiff, which is commonly used for generating difference files in LaTeX. We are confident that the final version of the manuscript, submitted for publication, is devoid of these formatting discrepancies. We kindly request the reviewer to consider this context, as finding a workaround for latexdiff's limitations would require additional effort that may not be warranted.